# Generating Likely Counterfactuals Using Sum-Product Networks

**Jiří Němeček, Tomáš Pevný & Jakub Mareček**
Department of Computer Science
Faculty of Electrical Engineering, Czech Technical University
Karlovo náměstí 13, Praha 2, 121 35
{nemecek.jiri,pevnytom,jakub.marecek}@fel.cvut.cz

## Abstract

The need to explain decisions made by AI systems is driven by both recent regulation and user demand. The decisions are often explainable only post hoc. In counterfactual explanations, one may ask what constitutes the best counterfactual explanation. Clearly, multiple criteria must be taken into account, although "distance from the sample" is a key criterion. Recent methods that consider the plausibility of a counterfactual seem to sacrifice this original objective. Here, we present a system that provides high-likelihood explanations that are, at the same time, close and sparse. We show that the search for the most likely explanations satisfying many common desiderata for counterfactual explanations can be modeled using Mixed-Integer Optimization (MIO). We use a Sum-Product Network (SPN) to estimate the likelihood of a counterfactual. To achieve that, we propose an MIO formulation of an SPN, which can be of independent interest. The source code with examples is available at https://github.com/Epanemu/LiCE.

## 1 Introduction

A better understanding of deployed AI models is needed, especially in high-risk scenarios (Dwivedi et al., 2023). Trustworthy and explainable AI (XAI) is concerned with techniques that help people understand, manage, and improve trust in AI models (Gunning et al., 2021; Burkart & Huber, 2021; Bodria et al., 2023). Explanations also serve an important role in debugging models to ensure that they do not rely on spurious correlations and traces of processing correlated with labels, such as timestamps. In a *post-hoc* explanation, a vendor of an AI system provides an individual user with a personalized explanation of an individual decision made by the AI system, improving the model's trustworthiness (Karimi et al., 2022; Li et al., 2023). In this context, personalized explanations are often called local explanations because they explain the model's decision locally, around a given sample, such as one person's input. Thus, local explanations provide information relevant to the user without revealing global information about the model, regardless of whether the model is interpretable *a priori*.

Consider, for example, credit decision-making in financial services. The models utilized need to be interpretable *a priori*, cf. the Equal Credit Opportunity Act in the US (ECOA) and related regulation (European Commission, 2016a;b) in the European Union, but an individual who is denied credit may still be interested in a personalized, local explanation. A well-known example of local explanations is the counterfactual explanation (CE). CE answers the question "How should a sample be changed to obtain a different result?" (Wachter et al., 2017). In the example of credit decision-making, a denied client might ask what they should do to obtain the loan. The answer would take the form of a CE. For example, "Had you asked for half of the loan amount, your application would have been accepted". As illustrated, CE can be easily understood (Byrne, 2005; Guidotti, 2022). However, their usefulness is influenced by many factors (Guidotti, 2022), including validity, similarity, sparsity, actionability, and plausibility.

This work focuses on the plausibility of counterfactual explanations. Unfortunately, plausibility does not have a clear definition. The definition of Guidotti (2022) suggests that CE should not be an outlier and measures it as the mean distance to the data. A Local Outlier Factor is often used

Table 1: *Method comparison.* A check mark indicates that a given method claims to possess the given feature. The star symbol (*) means that the method is model-agnostic as long as the classifier can be expressed using MIO. Complex data means data with continuous, categorical, ordinal, and discrete contiguous values. Exogenous property means that a method can generate unseen data samples as CEs. Regarding actionability, C-CHVAE disregards the monotonicity of features, and DiCE claims to achieve actionability through diversity without any data-specific constraints in place. All methods require validity and optimize some notion of similarity.

| Method | | Plausibility | Sparsity | Actionability | Complex data | Model-agnostic | Exogenous |
|---|---|---|---|---|---|---|---|
| PROPLACE | (Jiang et al., 2024) | ✓ | | | | | ✓ |
| C-CHVAE | (Pawelczyk et al., 2020) | ✓ | | only immut. | | ✓ | ✓ |
| FACE | (Poyiadzi et al., 2020) | ✓ | | ✓ | ✓ | ✓ | |
| DiCE | (Mothilal et al., 2020) | | ✓ | ✓ | | | ✓ |
| PlaCE | (Artelt & Hammer, 2020) | ✓ | ✓ | | | | ✓ |
| DACE | (Kanamori et al., 2020) | ✓ | | ✓ | ✓ | ✓* | ✓ |
| LiCE | Proposed here (Section 5) | ✓ | ✓ | ✓ | ✓ | ✓* | ✓ |

(e.g., Kanamori et al., 2020), but this method is not invariant of the data size. Alternatively, Jiang et al. (2024) define a "plausible region" as a convex hull of $k$ nearest neighbors of the factual. However, this region can still contain outliers. One can also use generative models, such as Adversarial Random Forests (Dandl et al., 2024) that incorporate plausibility in the generation process. This approach relies on multi-objective heuristic optimization.

Many other methods consider estimating the likelihood of CEs as a proxy for plausibility. This approach aligns with the definition of CE not being an outlier since outliers will have a low likelihood. One such approach uses (Conditional) Variational Auto-Encoders (Jordan et al., 1998; Pawelczyk et al., 2020; Stevens et al., 2024) in likelihood estimation. This approach does not provide a good way to handle categorical inputs and does not provide an efficient way to compute the exact likelihood of a CE. Plausible CE (PlaCE) proposed in (Artelt & Hammer, 2020) uses Gaussian mixture models in the framework of convex optimization to maximize likelihood in CE generation. Its limitations are the inability to handle categorical features and non-linear classifiers. Another common way to estimate likelihood is Kernel Density Estimation (KDE), which shares the inability to handle categorical features well. KDE is utilized by, e.g., FACE (Poyiadzi et al., 2020), which can also return CEs only from the training set.

**Our Contribution** We propose *Likely Counterfactual Explanations* (LiCE) method, which optimizes plausibility in combination with other desiderata (see Table 1). LiCE uses Sum-Product Networks (SPNs) of Poon & Domingos (2011), which are state-of-the-art tractable models to estimate likelihood. They naturally handle categorical features. This work combines the tradition of tractable probabilistic models with mixed-integer formulations by formulating the former in the latter.

In particular, we propose:

- A mixed-integer formulation of a trained Sum-Product Network estimating log-likelihood.
- Sum-Product Network as a measure of plausibility of CE, which allows the integration of plausibility directly into the MIO formulation.
- LiCE method for the generation of CEs. An MIO model that can be constrained by or optimized with respect to common desiderata regarding CE generation.

The advantage of our approach can be illustrated with an example from the German Credit dataset (Hofmann, 1994). See Figure 1, where CEs produced by several methods considering the diversity or plausibility of CE are compared against the factual (white cross) in the plane, where the horizontal axis represents the amount of credit and where the vertical axis is the duration.

For example, C-CHVAE (Pawelczyk et al., 2020), VAE and FACE (Poyiadzi et al., 2020) suggest reducing the duration by one year or more. The most plausible explanation produced by DiCE (Mothilal et al., 2020) counter-intuitively suggests doubling the credit amount to obtain it. PROPLACE (Jiang et al., 2024) suggests decreasing the credit amount below a third of the original amount. All of these counterfactuals are quite distant from the factual. In contrast, MIO finds a counterfactual with the sought credit amount and suggests decreasing the loan duration by a single

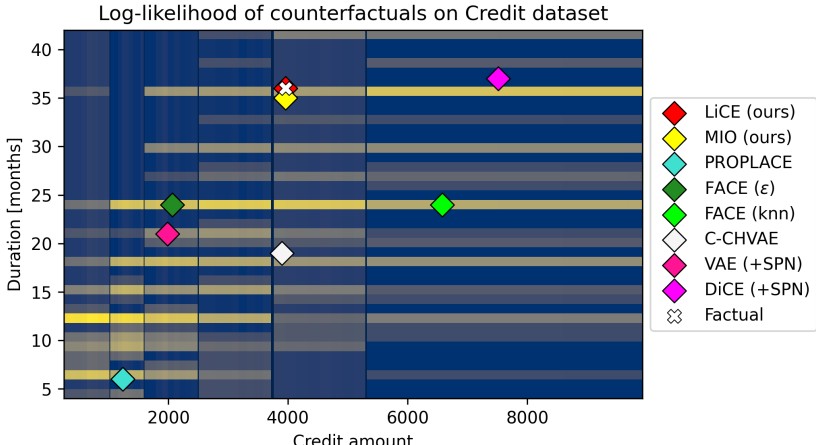

Figure 1: The heatmap shows the marginalized log-likelihood distribution of the German Credit dataset into a 2-dimensional space of Credit amount and Duration features, with extremely low values clipped to $-12.5$ for visual clarity. The factual (white cross) and CEs are also projected to the two dimensions. The factual is classified as being denied. Most CE methods choose distant points, sometimes with poor likelihood. The proposed method (LiCE) strikes a balance between likelihood and proximity.

month. Because the visualization is a 2-dimensional projection, some changes are not visualized. LiCE changes only one "hidden" feature. All other methods change at least five features (except MIO, which changes three), showing poor sparsity.

This example illustrates the issue of considering plausibility exclusively. High plausibility should ensure that the counterfactual is not an outlier, i.e., it is "realizable" by the client. However, this can lead to non-sparse, distant CEs, which are nonetheless difficult to realize.

**Notation used**  Throughout the paper, we consider a classification problem in which the dataset $\mathcal{D}$ is a set of 2-tuples $(\mathbf{x}, y) \in \mathcal{D}$. Each input vector $\mathbf{x} \in \mathcal{X} \subseteq \mathbb{R}^P$ consists of $P$ features and is taken from the input space $\mathcal{X}$ that can be smaller than $P$-dimensional real space (e.g., can contain categorical values). $x_j$ is the value of the $j$-th feature of the sample $\mathbf{x}$. We have $C$ classes and describe the set of classes $[C] = \{1, \ldots, C\}$. $y \in [C]$ is the true class of the sample $\mathbf{x}$. Finally, we have a classifier $h(\mathbf{x}) = \hat{y} \in [C]$ that predicts the class $\hat{y}$ for the sample $\mathbf{x}$. More details on the notation are given in Section A.

## 2 PREREQUISITES

### 2.1 COUNTERFACTUAL EXPLANATIONS

We define a counterfactual explanation in accordance with previous works as $\mathbf{x}' \in \mathcal{X}$ such that $h(\mathbf{x}) \neq h(\mathbf{x}')$ and the distance between $\mathbf{x}'$ and $\mathbf{x}$ is in some sense minimal (Guidotti, 2022; Wachter et al., 2017). We refer to $\mathbf{x}$ as factual and $\mathbf{x}'$ as counterfactual or CE. As mentioned above, there are many desiderata regarding the properties of CEs. Following Guidotti (2022), the common desiderata in which we are interested are:

- *Validity.* $\mathbf{x}'$ should be classified differently than $\mathbf{x}$
- *Similarity.* $\mathbf{x}'$ should be similar (close) to $\mathbf{x}$
- *Sparsity.* $\mathbf{x}'$ should change only a few features compared to $\mathbf{x}$, i.e., minimize $\|\mathbf{x}' - \mathbf{x}\|_0$
- *Actionability.* A counterfactual should not change features that cannot be changed (immutability). This includes the monotonicity of some features, e.g., age can only increase.
- *Plausibility.* CEs should have a high likelihood (be plausible) with respect to the distribution that has generated the dataset $\mathcal{D}$. This is sometimes interpreted as not being an outlier.

Guidotti (2022) describes also other desiderata, which we discuss in Section B.1

## 2.2 MIXED-INTEGER OPTIMIZATION

Mixed-Integer Optimization (MIO, (Wolsey, 2020)) is a powerful framework for modeling and solving optimization problems, where some decision variables take values from a discrete set while others are continuously valued. Non-trivially, the problem is in NP (Papadimitriou, 1981) and is NP-Hard, in general. There has been fascinating progress in the field in the past half-century (Bixby, 2012). State-of-the-art solvers based on the branch-and-bound-and-cut approach can often find global, certified optima for instances with millions of binary variables within hours, while there are pathological instances on under a thousand variables whose global optima are still unknown. Naturally, MIO is widely used in those areas of machine learning where both discrete and continuous decision variables need to be optimized jointly (e.g., Huchette et al., 2023). We use the more general abbreviation MIO, though we consider only mixed-integer *linear* formulations.

A crucial advance has been the mixed polytope formulation of Russell (2019), which neatly combines categorical and continuous values. A feature $j$ takes a continuous value from the range $[L_j, U_j]$ or one of the $K_j$ distinct categorical values. This is useful for modeling data with missing values, especially when there is a description of why the value is missing (Russell, 2019). To model the mixed polytope (Russell, 2019) of a counterfactual for the feature $j$, we create a one-hot encoding for $K_j$ discrete values into binary variables $d_{j,k}$ and a continuous variable $c_j$ with a binary indicator variable $d_j^{\text{cont}}$ equal to 1 when the feature takes a continuous value. In summary:

$$\sum_{k=1}^{K_j} d_{j,k} + d_j^{\text{cont}} = 1 \tag{1}$$

$$c_j = F_j d_j^{\text{cont}} - l_j + u_j \tag{2}$$

$$0 \le l_j \le (F_j - L_j) d_j^{\text{cont}} \tag{3}$$

$$0 \le u_j \le (U_j - F_j) d_j^{\text{cont}} \tag{4}$$

$$d_j^{\text{cont}}, d_{j,k} \in \{0, 1\} \qquad \forall k \in [K_j], \tag{5}$$

where $F_j$ is either the original value $x_j$ or the median value of continuous data of the mixed feature $j$ if the factual $x_j$ has one of the categorical values instead. Constraint (2) fixes the value of $c_j$ using two non-negative variables, $l_j$ and $u_j$, representing the decrease and increase in the continuous value, respectively. This construction facilitates the computation of the absolute difference from the factual. Since we minimize their (weighted) sum, at least one of them will always equal 0 (Russell, 2019).

## 2.3 SUM-PRODUCT NETWORKS

Probabilistic circuits (PCs) (Choi et al., 2020) are tractable probabilistic models (or rather, computational graphs) that support exact probabilistic inference and marginalization in time linear w.r.t. their representation size. Probabilistic circuits are defined by a tuple $(\mathcal{G}, \psi, \theta)$, where $\mathcal{G} = (\mathcal{V}, \mathcal{E})$ is a Directed Acyclic Graph (DAG) defining the computation model, a scope function $\psi : \mathcal{V} \to 2^{[P]}$ defines a subset of features over which the node defines its distribution, and a set of parameters $\theta$. The root node $n^{\text{root}}$ (a node without parents) has the scope function equal to all features, i.e., $\psi(n^{\text{root}}) = [P]$. To simplify the notation, we define a function $\text{pred}(n)$, giving a set of children (predecessors) of an inner node $n$ and denote $x_{\psi(n)}$ the features of $\mathbf{x}$ within the scope of $n$.

An important subclass of PCs is Sum-Product Networks (SPNs), which restrict PCs such that the inner (non-leaf) nodes are either sum nodes ($\mathcal{V}^\Sigma$) or product nodes ($\mathcal{V}^\Pi$).

*Leaf node* $n^{\text{L}} \in \mathcal{V}^{\text{L}} = \{n \,|\, \text{pred}(n) = \emptyset\}$ within SPNs takes a value $O_{n^{\text{L}}}$ from a (tractable) distribution over its scope $\psi(n^{\text{L}})$ parametrized by $\theta_{n^{\text{L}}}$.

*Product node* $n^\Pi \in \mathcal{V}^\Pi$ performs a product of probability distributions defined by its children

$$O_{n^\Pi}(x_{\psi(n)}) = \prod_{a \in \text{pred}(n^\Pi)} O_a(x_{\psi(a)}). \tag{6}$$

The scope of product nodes must satisfy decomposability, meaning that the scopes of its children are disjoint, i.e., $\bigcap_{a \in \mathrm{pred}(n^\Pi)} \psi(a) = \emptyset$, but complete $\bigcup_{a \in \mathrm{pred}(n^\Pi)} \psi(a) = \psi(n^\Pi)$.

*Sum node* $n^\Sigma \in \mathcal{V}^\Sigma$ has its value defined as

$$O_{n^\Sigma}(x_{\psi(n)}) = \sum_{a \in \mathrm{pred}(n^\Sigma)} w_{a,n^\Sigma} \cdot O_a(x_{\psi(a)}), \tag{7}$$

where weights $w_{a,n^\Sigma} \geq 0$ and $\sum_{a \in \mathrm{pred}(n^\Sigma)} w_{a,n^\Sigma} = 1$. The value of a sum node is thus a mixture of distributions defined by its children. The scope of each sum node must satisfy completeness (smoothness), i.e., it must hold that $\psi(a_1) = \psi(a_2) \; \forall a_1, a_2 \in \mathrm{pred}(n^\Sigma)$.

## 3 RELATED WORK

Pioneering work on counterfactual explanations (under the name "optimal action extraction") by Cui et al. (2015) considered classifiers based on additive tree models and extracted an optimal plan to change a given input to a desired class at a minimum cost using MIO. In parallel, similar approaches have been developed under the banner of "actionable recourse" (Ustun et al., 2019) or "algorithmic recourse" (Karimi et al., 2022; 2021). Developing upon this, Karimi et al. (2021) distinguish between contrasting explanations and consequential explanations, where actions are modeled explicitly in a causal model. Raimundo et al. (2024) coined a broader term "counterfactual antecedents". We use the term counterfactual explanations (CEs), popularized by, e.g., Wachter et al. (2017).

There is a plethora of work on the search for CEs, as recently surveyed (e.g., Karimi et al., 2022; Burkart & Huber, 2021; Guidotti, 2022; Bodria et al., 2023; Laugel et al., 2023). Below, we focus on methods for local CEs, with objectives related to the plausibility of CEs.

**DACE** (Kanamori et al., 2020) utilizes an MIO formulation, minimizing a combination of $\ell_1$-norm based Mahalanobis' distance and 1-Local Outlier Factor (1-LOF) for plausibility. The use of 1-LOF requires the use of $\mathcal{O}(|\mathcal{D}|)$ variables and $\mathcal{O}(|\mathcal{D}|^2)$ constraints. We improve on DACE by formulating the SPN as MIO to compute the likelihood. Thus, the number of variables and constraints does not depend on the dataset size but on the size of the SPN. Moreover, our flexible formulation allows us to maximize plausibility or constrain the CE not to be an outlier, similar to DACE. **PROPLACE** (Jiang et al., 2024) is also an MIO-based method for finding robust CEs within a "plausible region". The region is constructed as a convex hull of the factual and its (robust) nearest neighbors. The neighbors can, however, be outliers if a factual is not in a dense region of the data. Therefore, this approach is not faithful to the plausibility definition we use (Section 2.1). **FACE** (Poyiadzi et al., 2020) selects a CE from the training set $\mathcal{D}$, rather than generating it from $\mathcal{X}$. It works by navigating a graph of samples $\mathbf{x} \in \mathcal{D}$, where an edge exists between two samples if they are close or by connecting $k$-nearest neighbors. It further requires that a sample has density (evaluated by KDE) above a certain threshold. This approach is limited by the inability to generate exogenous CEs, which is not the case for our method. Del Ser et al. (2024) evaluate plausibility using the discriminator from a trained Generative Adversarial Network without estimating the data distribution directly.

However, similarly to LiCE, some works estimate the data distribution via a probabilistic model, such as Variational Auto-Encoder (VAE, Mahajan et al. (2020)). **C-CHVAE** (Pawelczyk et al., 2020) uses a Conditional VAE to search for plausible (they use the term *faithful*) CEs without the need of a metric in the original space. However, VAE provides only a lower bound on likelihood, and the solution lacks any guarantees of optimality. **PlaCE** (Artelt & Hammer, 2020) uses a Gaussian Mixture Model (GMM) to represent the data distribution. Their formulation approximates a GMM by a quadratic term and uses a general convex optimization solver. However, GMMs cannot handle categorical features, which are frequent in datasets of interest. More recently, **CeFlow** (Duong et al., 2023), **PPCEF** (Wielopolski et al., 2024) and Dombrowski et al. (2024) use Normalizing Flow Models, which, like GMMs, are non-trivial to formulate using linear constraints.

LiCE uses SPNs, probabilistic models that are tractable, naturally handle categorical features, and can have linear MIO formulation. SPNs are a strict generalization of GMMs (Aden-Ali & Ashtiani, 2020). We refer to Appendix G.3 for further discussion on using SPNs. Furthermore, while some research showed users' preference for counterfactual similarity in favor of plausibility (Kuhl et al., 2022), LiCE allows to balance these objectives using a single parameter.

## 4 MIXED-INTEGER FORMULATION OF SPN

Our contribution is built on a novel formulation of likelihood estimates provided by a Sum-Product Network (SPN) in Mixed-Integer Optimization (MIO). Eventually, this makes it possible to utilize the estimate to ensure plausible counterfactuals generated using MIO. Specifically, we propose an MIO formulation for a log space variant of a *fitted* SPN (Poon & Domingos, 2011) with fixed parameters. We perform all computations in log space because it enables formulation of product nodes with linear constraints, while sum nodes can be well approximated. In addition, it makes optimization less prone to numerical instabilities.

Let us introduce the MIO formulation following the definition of SPN in Section 2.3:

**Leaf nodes**  In any SPN, the leaves are represented by probability distributions over a single feature. In the case of discrete random variables, we can utilize the indicator $d_{j,k}$ that feature $j$ has value $k$ in the one-hot encoding. In the case of continuous random variables, we can utilize histogram approximations, that is, piecewise linear functions, whose mixed-integer formulations have been studied in considerable detail (cf. Huchette & Vielma, 2023). We suggest and utilize an alternative formulation of a histogram described in Section C.2.

**Product nodes**  In any SPN, each node $n$ combines the outputs $o_a$, $a \in \mathrm{pred}(n)$ of its predecessors. Consider now a product node $n \in \mathcal{V}^\Pi$, with output defined as a product of predecessor outputs. Since we consider all computations in log space, this translates to

$$o_n = \sum_{a \in \mathrm{pred}(n)} o_a \quad \forall n \in \mathcal{V}^\Pi. \tag{8}$$

**Sum nodes**  A sum node $n \in \mathcal{V}^\Sigma$ is defined as a weighted sum of predecessor $a$ outputs. In log space, the sum would translate to $o_n^* = \log \sum_{a \in \mathrm{pred}(n)} w_{a,n} \exp(o_a)$, which we cannot easily formulate as a linear expression. Considering $w_{a,n} \exp(o_a) = \exp(o_a + \log w_{a,n})$, we can approximate $\log \sum \exp(z)$ by $\max z$. Specifically, let $z_a = o_a + \log w_{a,n}$ and we bound

$$\max_{a \in \mathrm{pred}(n)} z_a = \log \exp(\max_{a \in \mathrm{pred}(n)} z_a)$$

$$\leq \log \sum_{a \in \mathrm{pred}(n)} \exp(z_a) = o_n^*$$

$$\leq \log \left( |\mathrm{pred}(n)| \exp(\max_{a \in \mathrm{pred}(n)} z_a) \right) = \log(|\mathrm{pred}(n)|) + \max_{a \in \mathrm{pred}(n)} z_a.$$

In other words, the approximate value $o_n$ of a sum node $n$ can be bound by the true value $o_n^*$ as

$$o_n^* - \log(|\mathrm{pred}(n)|) \ \leq \ o_n = \max_{a \in \mathrm{pred}(n)} z_a \ \leq \ o_n^*,$$

meaning that our approximation is a lower bound of the true $o_n^*$, and the error in the estimate is at the most logarithm of the number of predecessors. If we wanted an upper bound, we could easily add $\log(|\mathrm{pred}(n)|)$ to the value $o_n$. To formulate the max function, we can linearize it by introducing slack binary indicators $m_{a,n} \in \{0,1\}$ for each predecessor $a$ of sum node $n$

$$o_n \leq o_a + \log w_{a,n} + m_{a,n} \cdot T_n^{\mathrm{LL}} \quad \forall n \in \mathcal{V}^\Sigma, \forall a \in \mathrm{pred}(n) \tag{9}$$

$$\sum_{a \in \mathrm{pred}(n)} m_{a,n} = |\mathrm{pred}(n)| - 1 \quad \forall n \in \mathcal{V}^\Sigma \tag{10}$$

where $T_n^{\mathrm{LL}}$ is a big enough "big-M" constant (Wolsey, 2020). Constraint (10) ensures that constraint (9) is tight ($o_n \leq z_a$) for a single predecessor $a$ for which $m_a = 0$. Since we maximize the likelihood, the value of $o_n$ will be equal to $\max_a z_a$.

## 5 LIKELY COUNTERFACTUAL EXPLANATIONS

As our main contribution, we present a novel formulation for Likely Counterfactual Explanations (LiCE), which finds plausible CEs (with high likelihood) while satisfying common desiderata. Since

the optimization problem is written as MIO, the solution (CE) satisfies all constraints and is globally optimal. Throughout the section, we assume that all continuous values are scaled to the range $[0, 1]$.

We now describe how we formulate the input encoding, classification model, and various desiderata as MIO constraints. The potential of MIO to formulate similar constraints is well discussed in the literature (e.g., Russell, 2019; Kanamori et al., 2020; Mohammadi et al., 2021; Jiang et al., 2024), although the discussion rarely contains concrete formulations (Parmentier & Vidal, 2021). We discuss MIO formulations of the desiderata specific for the mixed polytope input encoding (Russell, 2019) in Section B.3.

**Input encoding**    To encode the input vector, we utilize the mixed polytope formulation (Russell, 2019), as explained in Eqs. 1–5 on page 4. The mixed polytope encoding works for purely continuous values by setting $K_j = 0$. For fully categorical features, one must disregard the variable $d_j^{\text{cont}}$ as explained in more detail in Section C.1.

The input to the classification model (and to the SPN) is then a set of all variables $c_j$ and $d_{j,k}$ (but not $d_j^{\text{cont}}$) concatenated into a single vector. With some abuse of the notation, we denote this vector $\mathbf{x}'$. When there is no risk of confusion, we denote the space of encoded inputs as $\mathcal{X}$ and the number of features after encoding as $P$.

**Model formulation**    We encode the classification model using the OMLT library (Ceccon et al., 2022), which simplifies the formulation of various ML models, although we focus on Neural Networks. Linear combinations in layers are modeled directly, while ReLUs are modeled using big-M formulations, though other formulations are possible (Fischetti & Jo, 2018).

**Validity**    Let $h^{\text{raw}} : \mathcal{X} \to \mathcal{Z}$ be the neural network model $h(\cdot)$ without activation at the output layer. Let $h^{\text{raw}}(\mathbf{x}')$ be the result obtained from the model implementation. Assuming that we have a binary classification task ($C = 2$), a neural network typically has a single output neuron ($\mathcal{Z} = \mathbb{R}$). A sample $\mathbf{x}$ is classified based on whether the raw output is above or below 0, i.e., $h(\mathbf{x}) = \mathbb{1}\{h^{\text{raw}}(\mathbf{x}) \geq 0\}$. Thus, depending on whether the factual is classified as 0 or 1, we set

$$h^{\text{raw}}(\mathbf{x}') \geq \tau \text{ or } h^{\text{raw}}(\mathbf{x}') \leq -\tau, \tag{11}$$

respectively, where $\tau \geq 0$ is a margin that can be set to ensure a higher certainty of the decision, improving the reliability of the CE. We present further formulations of the validity for $C > 2$ in Section B.3.1.

**Similarity and Sparsity**    To ensure similarity of the counterfactual, we follow Wachter et al. (2017) and Russell (2019) and use the somewhat non-standard $\|\cdot\|_{1,\text{MAD}}$ norm, weighed by inverse Median Absolute Deviation (MAD)

$$\|\mathbf{x}\|_{1,\text{MAD}} = \sum_{j=1}^{P} \left| \frac{x_j}{\text{MAD}_j} \right| \tag{12}$$

$$\text{MAD}_j = \text{median}_{(\mathbf{x},\cdot) \in \mathcal{D}} \left( |x_j - \text{median}_{(\mathbf{x},\cdot) \in \mathcal{D}}(x_j)| \right).$$

This metric also improves sparsity and adds scale invariance that is robust to outliers (Russell, 2019).

**Actionability**    We call a CE actionable if it satisfies monotonicity and immutability constraints. For immutability, the constraint is simply $x_j = x'_j$ for each immutable feature $j$. We can also set the input value as a parameter instead of a variable, omitting the feature encoding. Modeling monotonicity, i.e., that a given value cannot decrease/increase, is done using a single inequality for continuous features, e.g., $l_j = 0$ for a non-decreasing feature. For ordinal values, we fix to zero all one-hot dimensions representing smaller ordinal values, for non-decreasing features. Similarly, we can enforce basic causality constraints. Details are provided in Section B.3.3.

**Plausibility**    As explained in Section 4, fixed SPN fitted on the data allows us to estimate likelihood within MIO formulation. Negative likelihood can be added to the minimization objective with some multiplicative coefficient $\alpha > 0$. Alternatively, the likelihood can be used in constraints to force all generated CEs to have likelihood above a certain threshold $\delta^{\text{SPN}}$. Such constraint is simply

$$o_{n^{\text{root}}} \geq \delta^{\text{SPN}}, \tag{13}$$

where $\delta^{\text{SPN}}$ is a hyperparameter of our method, and $o_{n^{\text{root}}}$ is the likelihood estimated by the SPN.

**Full LiCE model**   In summary, our method optimizes the following problem:

$$\arg\min_{\mathbf{l},\mathbf{u},\mathbf{d}} (\mathbf{l}+\mathbf{u})^{\mathsf{T}}\mathbf{v}^{\mathrm{cont}} + (\mathbf{d}-\mathbf{d}^{\mathrm{fact}})^{\mathsf{T}}\mathbf{v}^{\mathrm{bin}} - \alpha \cdot o_{n^{\mathrm{root}}} \tag{14}$$

$$\text{s.t. mixed polytope conditions (1–5) hold}$$
$$\text{ML classifier constraints hold}$$
$$\text{validity constraints (e.g., 11) hold}$$
$$\text{SPN constraints (8–10) hold}$$
$$\text{plausibility constraint (13) holds}$$
$$\text{data-specific desiderata (e.g., actionability) constraints hold,}$$

where $\alpha$ is a parameter of LiCE, weighing the influence of log-likelihood in the objective, $\mathbf{l}$, $\mathbf{u}$ and $\mathbf{d}$ represent the vectors obtained by concatenation of the parameters in Eqs. 1–5. The vector $\mathbf{d}^{\mathrm{fact}}$ is the vector of binary variables of the encoded factual $\mathbf{x}$. $\mathbf{v}^{\mathrm{cont}}$ and $\mathbf{v}^{\mathrm{bin}}$ represent weights for continuous and binary variables, respectively. The weights for feature $j$ are $1/\mathrm{MAD}_j$ and thus Eq. 14 (when $\alpha = 0$) correspond to Eq. 12. Details about data-specific constraints are in Section E.1.

## 6   EXPERIMENTS

We first train a basic feed-forward Neural Network (NN) classifier with 2 hidden layers with ReLU activations. One could easily use one of the variety of ML models that can be formulated using MIO, including linear models, (gradient-boosted) trees, forests, or graph neural networks.

Secondly, we train an SPN to model the likelihood on the same training dataset. We include the class $y$ of a sample $\mathbf{x}$ in the training since we have prior knowledge of the counterfactual class. SPNs have a variety of training methods (Xia et al., 2023), of which we use a variant of LearnSPN (Gens & Domingos, 2013) implemented in the SPFlow library (Molina et al., 2019), though newer methods exist (e.g., Trapp et al., 2019).

**Data**   We tested on the Give Me Some Credit (GMSC) dataset (Fusion & Cukierski, 2011), the Adult dataset (Becker & Kohavi, 1996) and the German Credit (referred to as Credit) dataset (Hofmann, 1994). We dropped some outlier data and some less informative features (details in Section D) and performed all experiments in a 5-fold cross-validation setting.

**LiCE variants**   The main proposed model directly reflects the formulation (14). We compare two variants, one with a lower-bound on the log-likelihood ($\delta^{\mathrm{SPN}}$) at the median log-likelihood value of training samples, similar to Artelt & Hammer (2020). We also set $\alpha = 0$, to minimize purely the distance to factual. We refer to this as **LiCE (median)**. The other variant, **LiCE (optimize)**, is the opposite, i.e., we optimize a combination of distance and likelihood with $\alpha = 0.1$ and relax the plausibility constraint (Eq. 13). **MIO** represents our method without the SPN model directly formulated. We use all constraints described in Section 5, without the plausibility and SPN constraints. We use the SPN post hoc to select the most likely explanation.

MIO and LiCE are implemented using the open-source Pyomo modeling library (Bynum et al., 2021) that allows for the simple use of (almost) any MIO solver. We use the Gurobi solver (Gurobi Optimization, LLC, 2024). We solve each formulation for up to 2 minutes, after which we recover (up to) 10 best solutions. The entire implementation, together with the data, is available at `https://github.com/Epanemu/LiCE`.

**Compared methods**   We compare our methods to the **C-CHVAE** (Pawelczyk et al., 2020), **FACE** (Poyiadzi et al., 2020) and **PROPLACE** (Jiang et al., 2024) methods described in Section 3. We use the implementations of FACE and C-CHVAE provided in the CARLA library (Pawelczyk et al., 2021). We run FACE in two variants, connecting samples within a given distance ($\epsilon$) or by nearest neighbors (knn). For PROPLACE, we use the official implementation (Jiang et al., 2024). We omit PlaCE and DACE since their implementation does not support CE generation for Neural Networks.

In addition to those, we also compare to **DiCE** (Mothilal et al., 2020), a well-known method that focuses on generating a diverse set of counterfactuals. **VAE** is a method using a Variational Auto-Encoder. It is an implementation available in version 0.4 of the DiCE library based on the work of Mahajan et al. (2020). For DiCE and VAE, we select the most likely CE out of 10 generated CEs.

Table 2: Approximation quality of the SPN. The first row shows the mean likelihood of CEs, evaluated by the SPN. The second row is the mean output of the MIO formulation of the same SPN. The third row shows the mean difference.

|  | GMSC | Adult | Credit |
|---|---|---|---|
| True SPN output | $-25.62 \pm 4.64$ | $-18.15 \pm 3.89$ | $-28.79 \pm 3.28$ |
| MIO formulation output ($o_{n^{\text{root}}}$) | $-25.71 \pm 4.71$ | $-18.67 \pm 4.10$ | $-29.01 \pm 3.34$ |
| **SPN approximation error** | $0.09 \pm 0.57$ | $0.53 \pm 0.47$ | $0.23 \pm 0.24$ |

Table 3: The proportion of factual instances for which a given method generated a valid or actionable counterfactual. Actionable CEs satisfy the immutability and monotonicity of relevant features (see Section 2.1).

| | Method | **GMSC** (Fusion & Cukierski, 2011) | | **Adult** (Becker & Kohavi, 1996) | | **Credit** (Hofmann, 1994) | |
|---|---|---|---|---|---|---|---|
| | | Valid | Actionable | Valid | Actionable | Valid | Actionable |
| | **DiCE** (+spn) | **100.0%** | **100.0%** | 99.8% | 65.4% | 99.2% | 3.4% |
| | **VAE** (+spn) | 1.2% | 0.2% | 79.8% | 22.6% | 28.2% | 0.0% |
| | **C-CHVAE** | 84.8% | 21.2% | 13.4% | 9.0% | 11.6% | 9.6% |
| | **FACE** ($\epsilon$) | 98.8% | 15.8% | 63.2% | 31.2% | 28.0% | 10.4% |
| | **FACE** (knn) | 98.8% | 14.8% | 79.8% | 45.4% | 28.2% | 11.2% |
| | **PROPLACE** | 98.8% | 34.2% | 79.8% | 54.8% | 28.2% | 9.6% |
| ours | **MIO** (+spn) | **100.0%** | **100.0%** | **100.0%** | **100.0%** | **99.4%** | **99.4%** |
| | **LiCE** (optimize) | **100.0%** | **100.0%** | **100.0%** | **100.0%** | **99.4%** | **99.4%** |
| | **LiCE** (median) | 60.0% | 60.0% | 90.6% | 90.6% | **99.4%** | **99.4%** |

If a CE method requires any prior training, we use the default hyperparameters (or some reasonable values, details in Section E.2) and train it on the same training set. If a given method can take into account actionability constraints, we enforce them.

**Experimental settings** For all experiments, we assume that the SPN and NN are fitted and fixed. We generate CEs for 100 factuals using each method for each fold, summing up to 500 factuals per dataset. The factuals are randomly selected from both classes. Methods that can output more CEs (MIO, LiCE, DiCE, VAE) are set to find at most 10 CEs, and we select valid CE with the highest likelihood (evaluated by the SPN) post hoc. Further details on hyperparameters and experiment configurations are provided in Section E.

**Results** To assess the quality of the MIO approximation of the SPN, we compare the CE likelihood computed by the MIO solver and the true value computed by SPN in Table 2. The worst approximation error is at $0.53$ on average, which is just $2.92\%$. We find this surprisingly tight. Moreover, considering the differences between methods (cf. Table 4), this is acceptable.

The comparison of the CE methods is non-trivial since the factuals for which a given method successfully returned a valid counterfactual are not the same for all methods. See Table 3 for details on the success rate of the presented methods. For LiCE (median), the lower rates are caused by a failure to create a counterfactual candidate in time. For other methods, it is also a failure to follow the validity/actionability criteria, especially for the case where a valid CE exists but an actionable does not. Overall, these results show that MIO-based methods have a high success rate unless the constraints are too tight. Methods unconstrained by the likelihood, i.e., MIO and LiCE (optimize), have a 100% success rate, except for credit dataset, where a few ill-defined samples had no actionable counterfactual w.r.t. the NN. For our methods, all generated CEs are guaranteed to be both valid and actionable.

We now compare CE methods on plausibility, similarity, and sparsity measured by negative log-likelihood (evaluated by the SPN), $\|\cdot\|_{1,\text{MAD}}$, and by the number of modified features, respectively, see Table 4. The results are difficult to interpret since not every method produced a valid CE for each factual. However, MIO and LiCE have success rates among the highest (cf. Table 3) and still perform best not only with regards to likelihood but also in terms of similarity and sparsity. Results on a subset of factuals for which each method generated a valid CE paint a similar picture, see Table 11 in Section F.2.

Table 4: Mean negative log-likelihood (NLL), $\|\cdot\|_{1,\text{MAD}}$ distance, and the number of changed features, measured on *valid* generated counterfactuals, with information about standard deviation. The log-likelihood is estimated by the SPN. The number of valid counterfactuals generated by a given method varies (see Table 3), so the direct comparison between methods is non-trivial. The (+spn) means that the given method generates 10 CEs from which we choose the likeliest valid counterfactual using the SPN. For all measures, a lower value is better.

| Method | GMSC (Fusion & Cukierski, 2011) | | | Adult (Becker & Kohavi, 1996) | | | Credit (Hofmann, 1994) | | |
|---|---|---|---|---|---|---|---|---|---|
| | NLL ↓ | Similarity ↓ | Sparsity ↓ | NLL ↓ | Similarity ↓ | Sparsity ↓ | NLL ↓ | Similarity ↓ | Sparsity ↓ |
| DiCE (+spn) | $29.1 \pm 5.2$ | $27.3 \pm 6.7$ | $6.5 \pm 1.1$ | $21.0 \pm 3.0$ | $26.7 \pm 13.1$ | $4.5 \pm 1.8$ | $51.0 \pm 17.9$ | $27.7 \pm 7.2$ | $8.7 \pm 2.1$ |
| VAE (+spn) | $\mathbf{18.0 \pm 2.2}$ | $17.2 \pm 3.7$ | $8.0 \pm 1.2$ | $18.4 \pm 3.6$ | $37.1 \pm 13.2$ | $5.4 \pm 1.5$ | $48.5 \pm 17.1$ | $28.2 \pm 7.5$ | $10.8 \pm 1.9$ |
| C-CHVAE | $25.6 \pm 2.4$ | $18.0 \pm 4.7$ | $8.3 \pm 0.7$ | $18.0 \pm 3.4$ | $8.2 \pm 5.2$ | $2.8 \pm 0.9$ | $32.3 \pm 3.6$ | $12.9 \pm 4.8$ | $6.6 \pm 1.5$ |
| FACE ($\epsilon$) | $29.4 \pm 7.6$ | $14.9 \pm 4.0$ | $8.4 \pm 1.1$ | $14.9 \pm 2.9$ | $15.2 \pm 9.1$ | $3.8 \pm 1.3$ | $43.2 \pm 17.6$ | $18.4 \pm 6.0$ | $7.0 \pm 1.4$ |
| FACE (knn) | $29.0 \pm 7.7$ | $15.0 \pm 4.2$ | $8.4 \pm 1.1$ | $14.5 \pm 2.8$ | $14.3 \pm 7.8$ | $3.7 \pm 1.2$ | $44.2 \pm 17.5$ | $18.6 \pm 6.3$ | $7.1 \pm 1.5$ |
| PROPLACE | $27.9 \pm 4.3$ | $12.9 \pm 3.2$ | $6.4 \pm 1.2$ | $15.4 \pm 2.4$ | $23.2 \pm 8.6$ | $4.8 \pm 1.3$ | $38.4 \pm 15.2$ | $24.4 \pm 6.7$ | $9.0 \pm 1.3$ |
| MIO (+spn) | $27.9 \pm 6.6$ | $\mathbf{5.9 \pm 1.5}$ | $\mathbf{2.1 \pm 0.8}$ | $17.8 \pm 3.8$ | $5.8 \pm 3.8$ | $2.2 \pm 0.9$ | $43.6 \pm 17.5$ | $\mathbf{4.4 \pm 2.8}$ | $2.3 \pm 1.1$ |
| LiCE (optim.) | $25.6 \pm 4.6$ | $\mathbf{5.9 \pm 1.6}$ | $2.6 \pm 1.1$ | $18.1 \pm 3.9$ | $\mathbf{5.6 \pm 3.8}$ | $\mathbf{2.1 \pm 1.0}$ | $\mathbf{28.8 \pm 3.3}$ | $\mathbf{4.4 \pm 2.8}$ | $2.3 \pm 1.2$ |
| LiCE (median) | $18.3 \pm 2.2$ | $11.0 \pm 3.4$ | $4.4 \pm 1.2$ | $\mathbf{12.9 \pm 1.0}$ | $9.7 \pm 6.6$ | $3.0 \pm 1.4$ | $29.9 \pm 3.1$ | $\mathbf{4.4 \pm 2.9}$ | $\mathbf{2.1 \pm 1.2}$ |

The plausibility of LiCE (median), evaluated by the SPN, seems to be the best across datasets. Except for GMSC, where VAE is comparable (on very few factuals - 1.2%). Interestingly, the optimizing variant of LiCE achieves a better mean objective value on Adult and GMSC than the median variant, despite the median version's objective function not accounting for the NLL. This is in part because LiCE (optimize) has a bigger feasible space, allowing it to generate closer CEs with a likelihood worse than the median of the training set. The fact that LiCE (optimize) beats MIO on Adult in similarity (which MIO directly optimizes) is counterintuitive. It is caused by choosing the most likely CE out of a set of 10. This set includes local optima that are farther from the factual, but might have a higher likelihood.

Although for some datasets, the plausibility results are comparable between multiple methods, the similarity and sparsity remain dominated by our methods. We must also point out that merely adding the SPN as a post-hoc evaluation to some existing method (e.g., DiCE) performs significantly worse. Further comparisons and discussion of the results are in Sections F and G.

**Limitations**  Our method shares the limitations of all MIO methods with respect to scalability and computational complexity. The additional SPN formulation leads to some computational overhead, especially when using the likelihood threshold, as exemplified in the results on the GMSC dataset in Table 3.

Our method relies on an SPN to evaluate likelihood, i.e., plausibility. One may question the capability of an SPN to accurately model the data distribution. We empirically show a strong correlation between the SPN likelihood and the true probability in Section G.4. Furthermore, in Appendix F.4, we use synthetic data to show that the true probability of CEs generated by LiCE is comparable to the probability of CEs generated by other well-performing methods.

## 7 DISCUSSION AND CONCLUSIONS

We have presented a comprehensive method for generating counterfactual explanations called LiCE. In Section 5, we show that our method satisfies the most common desiderata–namely validity, similarity, sparsity, actionability and, most importantly, *plausibility*.

Our method shows promising performance at the intersection of plausibility, similarity, and sparsity. It also reliably generates high-quality, valid, and actionable CEs. However, time concerns are relevant once the full SPN is formulated within the model.

In future work, the limitations of using MIO could be addressed by approximation algorithms. Additionally, other SPN-based models (e.g., Trapp et al., 2020) could be considered to estimate plausibility. Last but not least, the MIO formulation of a Sum-Product Network can be of independent interest.

ACKNOWLEDGMENTS

This work has received funding from the European Union's Horizon Europe research and innovation programme under grant agreement No. 10107056. The contribution of Tomáš Pevný has been funded by the Czech Grant Agency under project 22-32620S. Finally, access to the computational infrastructure of the OP VVV funded project CZ.02.1.01/0.0/0.0/16_019/0000765 "Research Center for Informatics" is also gratefully acknowledged.

We thank anonymous reviewers for their insightful comments that led to the strengthening of this work.

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

## A  NOTATION

Generally, the notation follows these rules:

- Capital letters typically refer to amounts of something, as in classes, features, bins, etc. Exceptions are $U, L$, and $F$, which are taken from the original work (Russell, 2019).

Table 5: General functions used

| General function symbols | |
| --- | --- |
| $\lvert \cdot \rvert$ | Absolute value (if scalar) or size of the set |
| $[\cdot]$ | Set of integers, $[N] = \{1, 2, \ldots, N\}$ |
| $\mathbb{1}\{\cdot\}$ | Equal 1 if input is true, 0 otherwise |
| $\lVert \cdot \rVert_0$ | $\ell_0$ norm, number of non-zero elements |
| $2^{[P]}$ | Set of all subsets of $[P]$ |

Table 6: Symbols used as indices

| Indices | |
| --- | --- |
| $j$ | Index of features, typically $j \in [P]$ |
| $(m)$ | Index of counterfactuals within a set $\mathcal{C}_\mathbf{x}$, typically $m \in [M]$ |
| $n$ | A node of the SPN, $n \in \mathcal{V}$ |
| $i$ | Index of bins of a histogram in a leaf node $(n)$, typically $i \in [B_n]$ |
| $a$ | A predecessor node (of node $n$) in the SPN, usually $a \in \mathrm{pred}(n)$ |
| $k$ | A class ($k \in [C]$) or categorical value ($k \in [K_j]$) index |
| $e$ | Index of the feature that is changed as an effect of causal relation R |

- Caligraphic capital letters denote sets or continuous spaces.
- Small Latin letters are used as indices, variables, or parameters of the MIP formulation.
- Small Greek letters refer to hyperparameters of the LiCE formulation or parameters of the SPN (scope $\psi$, parameters $\theta$).
- Subscript is used to specify the position of a scalar value in a matrix or a vector. When in parentheses, it specifies the index of a vector within a set.
- Superscript letters refer to a specification of a symbol with otherwise intuitively similar meaning. Except for $\mathbb{R}^P$, where $P$ has the standard meaning of $P$-dimensional.
- A hat ($\hat{\ }$) symbol above an element means that the element is the output of the Neural Network $h(\cdot)$.
- A prime ($'$) symbol as a superscript of an element means that the element is a part of (or the output of) the counterfactual.
- In bold font are only vectors. When we work with a scalar value, the symbol is in regular font.

The specific meanings of symbols used in the article are shown in Tables 5 to 9. The symbols are divided into groups.

- Functions non-specific to our task (Table 5)
- Used indices (Table 6)
- LiCE (hyper)parameters that can be tuned (Table 7)
- Classification task and SPN symbols (Table 8)
- MIO formulation parameters and variables (Table 9)

# B  CE DESIDERATA

## B.1  OTHER DESIDERATA FOR CES

We present more desiderata by Guidotti (2022) that we consider.

- *Diversity.* Each $\mathbf{x}'_{(m)} \in \mathcal{C}_\mathbf{x}$ should be as different as possible from any other CE in the set, ideally by proposing changes in different features. For example, one CE recommends

Table 7: Input parameters into the LiCE formulation

| | LiCE (hyper)parameters |
|---|---|
| $\tau$ | The minimal difference between counterfactual class ($h^{\mathrm{raw}}(\mathbf{x}')_{\hat{y}'}$) and factual class ($h^{\mathrm{raw}}(\mathbf{x}')_{\hat{y}}$) NN output value. Alternatively, for binary classification, it is the requirement for a minimal absolute value of the NN output before sigmoid activation ($h^{\mathrm{raw}}(\mathbf{x}')$). |
| $\rho$ | Limit for the relative difference of values of the objective function within the set of closest counterfactuals $\mathcal{C}_{\mathbf{x}}$. |
| $\alpha$ | Weight of negative log-likelihood in the objective function |
| $\epsilon_j$ | Minimal change in continuous value $c_j$ of $j$-th feature. The absolute difference between $x'_j$ and $x_j$ is either 0, or at least $\epsilon_j$. |
| $\delta^{\mathrm{SPN}}$ | Lower bound on the estimated value of likelihood of the generated counterfactual. |

Table 8: Symbols of the classification task, CE search, and SPNs

| | Classification task symbols |
|---|---|
| $P$ | Number of features |
| $C$ | Number of classes |
| $\mathcal{D}$ | The dataset, set of 2-tuples $(\mathbf{x}, y) \in \mathcal{D}$ |
| $\mathcal{X}$ | Input space $\mathcal{X} \subseteq \mathbb{R}^P$ |
| $\mathbf{x}$ | A (factual) sample $\mathbf{x} \in \mathcal{X}$ |
| $x_j$ | A $j$-th feature of sample $\mathbf{x}$ |
| $y$ | Ground truth of sample $\mathbf{x}$, $y \in [C]$ |
| $h(\cdot)$ | Classifier we are explaining $h : \mathcal{X} \to [C]$ |
| $\hat{y}$ | Classifier-predicted class $h(\mathbf{x}) = \hat{y} \in [C]$ |
| $h^{\mathrm{raw}}(\cdot)$ | NN classifier output without activation $h^{\mathrm{raw}} : \mathcal{X} \to \mathcal{Z}$ |
| $\mathcal{Z}$ | Output space of the NN classifier, without sigmoid/softmax activation |
| | **Counterfactual generation symbols** |
| $\|\cdot\|_{1,\mathrm{MAD}}$ | Counterfactual distance function (see Eq. 12) |
| $\mathcal{C}_{\mathbf{x}}$ | Set of generated counterfactuals for factual $\mathbf{x}$ |
| $M$ | Number of sought counterfactuals, $M \geq |\mathcal{C}_{\mathbf{x}}|$ |
| $\mathbf{x}'$ | Counterfactual explanation of $\mathbf{x}$, $\mathbf{x}' \in \mathcal{C}_{\mathbf{x}}$ |
| $\mathbf{x}'^*$ | Optimal (closest) counterfactual |
| $\mathbf{x}'_{(m)}$ | $m$-th counterfactual explanation of factual $\mathbf{x}$ |
| $x'_j$ | A value of $j$-th feature of the counterfactual |
| $\hat{y}'$ | Predicted class of the counterfactual (can be a parameter of LiCE) |
| | **Sum Product Network symbols** |
| $\mathcal{V}$ | Set of nodes of the SPN |
| $\mathcal{V}^{\mathrm{L}}$ | Set of leaf nodes |
| $\mathcal{V}^{\Sigma}$ | Set of sum nodes |
| $\mathcal{V}^{\Pi}$ | Set of product nodes |
| $\mathrm{pred}(\cdot)$ | Function returning children (predecessors) of a node |
| $\psi(\cdot)$ | Scope function mapping nodes to their input features $\psi : \mathcal{V} \to 2^{[P]}$ |
| $\theta$ | Parameters of the SPN |
| $O_n$ | Output value of a node $n \in \mathcal{V}$ |
| $w_{a,n}$ | Weight of output value of predecessor node $a$ in computing the value of sum node $n$. |
| $n^{\mathrm{root}}$ | Root node, its value is the value of the SPN |

Table 9: Used variables and parameters in the MIO formulation

| **MIO formulation variables** | |
| --- | --- |
| $l_j$ | Decrease in continuous value of $j$-th feature. |
| $\mathbf{l}$ | Concatenated vector of all $l_j$. |
| $u_j$ | Increase in continuous value of $j$-th feature. |
| $\mathbf{u}$ | Concatenated vector of all $u_j$. |
| $c_j$ | Continuous value of $j$-th CE feature. |
| $d_{j,k}$ | 1 iff $x'_j$ takes $k$-th categorical value $k \in K_j$. |
| $\mathbf{d}$ | All variables $d_{j,k}$ concatenated into a vector. |
| $d_j^{\text{cont}}$ | 1 iff $x'_j$ takes continuous value $c_j$. |
| $h^{\text{raw}}(\cdot)_k$ | Value of $h^{\text{raw}}$, corresponding to class $k \in [C]$. |
| $g_k$ | 1 iff class $k \in [C]$ has higher $h^{\text{raw}}$ value than the factual class. |
| $s_j$ | 1 iff $j$-the feature changed, i.e., $x_j \neq x'_j$. |
| $r$ | 1 iff causal relation $R$ is activated, i.e., cause is satisfied and effect is enforced. |
| $\bar{b}_{n,i}$ | 1 iff $x'_j$ does *not* belong to the $i$-th bin ($i \in [B_n]$), assuming $j$-th feature corresponds to node $n$, i.e., $\psi(n) = \{j\}$. |
| $o_n$ | Estimated output value of SPN node $n \in \mathcal{V}$. |
| $m_{a,n}$ | Binary slack indicator for sum node $n \in \mathcal{V}^\Sigma$ equal to 0 if output of predecessor $a$ constrains output of $n$ tightly. |
| **MIO formulation parameters** | |
| $L_j$ | Lower bound on continuous values of $j$-th feature. In our implementation, equal to 0. |
| $U_j$ | Upper bound on continuous values of $j$-th feature. In our implementation, equal to 1. |
| $F_j$ | Default continuous value of $j$-th feature, equal to the value of the factual $x_j$, if it has continuous value. Otherwise equal to the median. |
| $K_j$ | Number of categorical values of $j$-th feature. |
| $f_j$ | Equal to $x_j$, if it has categorical value. If $x_j$ is continuous, $f_j$ is removed, and so are all constraints containing it. |
| $S$ | Maximal number of feature value changes of $\mathbf{x}'$ compared to $\mathbf{x}$. Sparsity limit. |
| $R$ | Example causal relation: if $j$-th feature increases, $e$-th feature must decrease. |
| $B_n$ | Number of bins in the histogram of leaf node $n$. |
| $t_{n,i}$ | Threshold between $i - 1$-th and $i$-th bin in histogram of leaf node $n$ |
| $q_{n,i}$ | Likelihood value of $i$-th bin of node $n$. |
| $\mathbf{v}^{\text{bin}}$ | Vector of respective $\|\cdot\|_{1,\text{MAD}}$ weights for binary one-hot encodings. |
| $\mathbf{v}^{\text{cont}}$ | Vector of respective $\|\cdot\|_{1,\text{MAD}}$ weights for continuous values. |
| $\mathbf{d}^{\text{fact}}$ | One-hot encoded vector of factual categorical values corresponding to $\mathbf{d}$. |
| $T_n^{\text{LL}}$ | A "big-M" constant for sum node $n$, used for slack in the computation of $\max$. |

increasing the income; another one should recommend decreasing the loan amount instead. An important example of a CE library aiming for diversity is DiCE (Mothilal et al., 2020). In MIO, this is usually achieved by adding constraints and resolving the formulation (Russell, 2019; Mohammadi et al., 2021).

- *Causality.* Given that we know some causal relationships between the features, the generated CEs should follow them. For example, if $\mathbf{x}'$ contains a decrease in the total loan amount, the number of payments or their amount should also decrease.

## B.2 Our approach to these desiderata

**Causality**  Like actionability, causality depends on prior knowledge of the data. In causality, the constraints are in the form of implications (Mahajan et al., 2020). We describe a way to model causal constraints where, if one value changes in a certain direction, then another feature must change accordingly. Details are provided in Section B.3.3.

**Diversity and Robustness**  The diversity of CEs has been recently surveyed by Laugel et al. (2023). Methods for CE diversity in the context of MIO exist (Russell, 2019; Mohammadi et al., 2021). While their approach can be applied to our model too, here we simply generate a set of top-$M$ counterfactuals closest to the global optimum. We can optionally limit the maximal distance relative to the optimal CE; see Section B.3.5. Regarding the robustness of the counterfactuals, Artelt et al. (2021) show that finding plausible CEs indirectly improves the robustness. Thus, we do not add any further constraints to the model despite this being a viable option (e.g., Maragno et al., 2024; Jiang et al., 2024).

## B.3 MIO formulations of Desiderata

The following MIO formulations of the desiderata are novel in that we came up with them, and, to the best of our knowledge, they were not formalized before. They are not too complex, but we formulate them for completeness.

### B.3.1 Validity

For $C > 2$ classes, the raw output has $C$ dimensions ($\mathcal{Z} = \mathbb{R}^C$), and the classifier assigns the class equal to the index of the highest value, i.e., $h(\mathbf{x}) = \arg\max_{k \in [C]} h^{\mathrm{raw}}(\mathbf{x})_k$. Let $\hat{y}'$ be the desired counterfactual class. The validity constraint, given that we specify the counterfactual class prior, is then

$$h^{\mathrm{raw}}(\mathbf{x}')_{\hat{y}'} - h^{\mathrm{raw}}(\mathbf{x}')_k \geq \tau \quad \forall k \in [C] \setminus \{\hat{y}'\}. \tag{15}$$

Note that we can also implement a version where we do not care about the counterfactual class $\hat{y}'$ in advance by the following

$$
\begin{aligned}
g_k = 1 &\implies h^{\mathrm{raw}}(\mathbf{x}')_k - h^{\mathrm{raw}}(\mathbf{x}')_{\hat{y}} \geq \tau \quad \forall k \in [C] \setminus \{\hat{y}\} \\
g_k = 0 &\implies h^{\mathrm{raw}}(\mathbf{x}')_k - h^{\mathrm{raw}}(\mathbf{x}')_{\hat{y}} \leq \tau \quad \forall k \in [C] \setminus \{\hat{y}\} \\
&\qquad\qquad \sum_{k \in [C] \setminus \{\hat{y}\}} g_k \geq 1,
\end{aligned}
\tag{16}
$$

where $\implies$ can be seen either as an indicator constraint or as an implication (Williams, 2013), $g_k$ is equal to 1 if and only if class $k$ has a higher value than the factual class $\hat{y}$ in the raw output. The sum then ensures that at least one other class has a higher value.

A wide variety of constraints ensuring validity are possible. For example, we can ensure that the factual class has the lowest score by setting $\sum_{k \in [C] \setminus \{\hat{y}\}} g_k \geq C - 1$, or we could enforce a custom order of classes.

### B.3.2 SPARSITY

To constrain the sparsity further, we can set an upper bound $S$ on the number of features changed

$$\sum_j s_j \leq S$$

$$
\begin{aligned}
s_j &\geq 1 - d_{j,f_j} & \forall j \in [P] \\
s_j &\geq d_{j,k} & \forall j \in [P], \, \forall k \in [K_j] \setminus \{f_j\} \\
s_j &\geq l_j + u_j & \forall j \in [P] \\
s_j &\in \{0,1\} & \forall j \in [P],
\end{aligned}
\tag{17}
$$

where we use the binary value $s_j$ that equals 1 if the $j$-th feature changed, the $f_j$ is the categorical value of attribute $j$ of the factual (if applicable).

Neither LiCE nor MIO use this constraint.

### B.3.3 CAUSALITY

Consider the following example of a causal relation $R$. If feature $j$ increases its value, another feature $e$ must decrease. For continuous ranges, this is formulated as

$$
\begin{aligned}
r &\geq u_j - l_j \\
l_e &\geq r\epsilon_e \\
u_e &\leq 1 - r \\
r &\in \{0,1\},
\end{aligned}
\tag{18}
$$

where $\epsilon_e$ is a minimal change in the value of feature $e$ and $r$ equals 1 if the relation $R$ is active. In the case when the features are ordinal, we can assume that their values are just variables representing categorical one-hot encoding, ordered by indices and use:

$$
\begin{aligned}
r &\geq \sum_{k=f_j+1}^{K_j} d_{j,k} \\
r &\leq \sum_{k=1}^{f_e} d_{e,k} \\
r &\in \{0,1\},
\end{aligned}
\tag{19}
$$

where $f_j$ is the categorical value of the factual in feature $j$. Naturally, one can see that we can use any combination of increasing/decreasing values in continuous and categorical feature spaces. With these formulations, we can also model monotone values, such as age or education. We simply replace the variable $r$ by 1.

One can formulate any directed graph composed of these causal relations by decomposing it into pairwise relations, one per edge. This way, we can encode commonly used Structural Causal Models that utilize directed graphs to express causality.

### B.3.4 COMPLEX DATA

We use the umbrella term "Complex data" for tabular data with non-real continuous values. This includes categorical (e.g., race), binary (e.g., migrant status), ordinal (e.g., education), and discrete contiguous (e.g., number of children) values.

For binary, we use a simple 0-1 encoding; categorical data is encoded into one-hot vectors; and discrete features are discretized by fixing their value to an integer variable within the formulation. Since we normalize all values to the $[0,1]$ range, we introduce a proxy integer variable $z_j$:

$$
\begin{aligned}
(F_j - l_j + u_j) \cdot \text{scale}_j + \text{shift}_j &= z_j \\
z_j &\in \mathbb{Z}
\end{aligned}
$$

For ordinal variables, we use the same encoding as categorical values, with the addition of the one-hot encoding being sorted by value rank to allow for the causality/monotonicity to be enforced.

### B.3.5 DIVERSITY

Instead of a single counterfactual, the solver returns (up to) $M$ counterfactuals closest to the global optimum, optionally within some distance range. This range is defined in terms of the objective function, which is the distance of a counterfactual in our case. In other words, we search for a set $\mathcal{C}_{\mathbf{x}} = \{\mathbf{x}'_{(1)}, \ldots, \mathbf{x}'_{(M)}\}$ of counterfactuals that have a similar distance to the factual.

Let $\mathbf{x}'^*$ be the closest CE satisfying all other constraints; we can set a parameter $\rho$ that represents the relative distance of all CEs to the $\mathbf{x}'^*$ leading to the generation of set

$$\mathcal{C}_{\mathbf{x}} = \{\mathbf{x}' \mid \|\mathbf{x} - \mathbf{x}'\|_{1,\text{MAD}} \leq (1 + \rho) \cdot \|\mathbf{x} - \mathbf{x}'^*\|_{1,\text{MAD}}\}.$$

Nevertheless, we disregard the relative distance parameter and search for the $M$ closest CEs. Later, we sift through the set $\mathcal{C}$ of top-$M$ counterfactuals, looking for the most likely CEs. Here, one could perform any filtering.

## C OTHER MIO FORMULATIONS

### C.1 MIXED POLYTOPE FORMULATION CORRECTION

For purely categorical features, the original mixed polytope (Russell, 2019) implementation contains an issue. The first categorical value (represented by zero) is mapped to the continuous variable. This seems to work fine for the logarithmic regression (Russell, 2019), but it failed on non-monotone neural networks, leading to non-binary outputs. This was corrected by replacing the continuous variable $c_j$ with another binary decision variable, making it a standard one-hot encoding.

### C.2 SPN HISTOGRAM FORMULATION

In practice, the probability distribution of a leaf $n \in \mathcal{V}^{\text{L}}$ trained on data is a histogram on a single feature $j$, i.e., $\psi(n) = \{j\}$. The interval of possible values of $x'_j$ is split into $B_n$ bins, delimited by $B_n + 1$ breakpoints denoted $t_{n,i}$, $i \in [B_n + 1]$.

Because modeling that a value of a variable belongs to a union of intervals is simpler than an intersection, we consider variables $\bar{b}_{n,i}$ that equal 1 if and only if the value $x'_j$ does *not* belong to the interval $[t_{n,i}, t_{n,i+1})$. This leads to a set of constraints

$$\bar{b}_{n,i} \geq t_{n,i} - x'_j \qquad\qquad \forall n \in \mathcal{V}^{\text{L}}, \forall i \in [B_n] \qquad (20)$$

$$\bar{b}_{n,i} \geq x'_j + \epsilon_j - t_{n,i+1} \qquad\qquad \forall n \in \mathcal{V}^{\text{L}}, \forall i \in [B_n] \qquad (21)$$

$$\sum_{i=1}^{B_n} \bar{b}_{n,i} = B_n - 1 \qquad\qquad \forall n \in \mathcal{V}^{\text{L}} \qquad (22)$$

$$o_n = \sum_{i=1}^{B_n} (1 - \bar{b}_{n,i}) \log q_{n,i} \qquad\qquad \forall n \in \mathcal{V}^{\text{L}} \qquad (23)$$

$$\bar{b}_{n,i} \in \{0, 1\} \qquad\qquad \forall n \in \mathcal{V}^{\text{L}}, \forall i \in [B_n], \qquad (24)$$

where $q_{n,i}$ is the likelihood value in a bin $i$ and $o_n$ is the output value of the leaf node $n$. $\epsilon_j$ is again the minimal change in the feature $j$ and ensures that we consider an open interval on one side. We use the fact that all values $x_j$ (thus also $t_{n,i}$) are in the interval $[0, 1]$. Eq. 20 sets $\bar{b}_{n,i} = 1$ if $x'_j < t_{n,i}$ and Eq. 21 sets $\bar{b}_{n,i} = 1$ for values on the other side of the bin $x'_j \geq t_{n,i+1}$. Eq. 22 ensures that a single bin is chosen and Eq. 23 sets the output value to the log value of the bin that $\mathbf{x}'$ belongs to. This implementation of bin splitting is inspired by the formulation of interval splitting in piecewise function fitting of Goldberg et al. (2021).

We assume that the bins cover the entire space, which we can ensure by adding at most 2 bins on both sides of the interval.

# D DATA MODIFICATIONS

We remove samples with missing values. Optionally, we also remove some outlier data or uninformative features.

**GMSC**   We do not remove any feature in GMSC, but we keep only data with reasonable values to avoid numerical issues within MIO. The thresholds for keeping the sample are as follows

- `MonthlyIncome` $< 50000$
- `RevolvingUtilizationOfUnsecuredLines` $< 1$
- `NumberOfTime30-59DaysPastDueNotWorse` $< 10$
- `DebtRatio` $< 2$
- `NumberOfOpenCreditLinesAndLoans` $< 40$
- `NumberOfTimes90DaysLate` $< 10$
- `NumberRealEstateLoansOrLines` $< 10$
- `NumberOfTime60-89DaysPastDueNotWorse` $< 10$
- `NumberOfDependents` $< 10$

this removes around 5.5% of data after data with missing values was removed. We could combat the same issues by taking a $\log$ of some of the features. In our "pruned" GMSC dataset, there are 113,595 samples and 10 features, none of which are categorical, 7 are discrete contiguous, and the remaining 3 are real continuous. Further details are in the preprocessing code.

**Adult**   In the Adult dataset, we remove 5 features

- `fnlwgt` which equals the estimated number of people the data sample represents in the census, and is thus not actionable and difficult to obtain for new data, making it less useful for predictions,
- `education-num` because it can be substituted by ordinal feature `education`,
- `native-country` because it is again not actionable, less informative, and also heavily imbalanced,
- `capital-gain` and `capital-loss` because they contain few non-zero values.

It is not uncommon to remove the features we did, as some of them also have many missing values. We remove only about 2% of the data by removing samples with missing values. We are left with 47,876 samples and 9 features, 5 of which are categorical, 1 is binary, 1 ordinal, and the remaining 2 are discrete contiguous. Further details are in the preprocessing code.

**Credit**   We do not remove any samples or features for the Credit dataset. The dataset contains 1,000 samples and 20 features, 10 of which are categorical, 2 are binary, 1 ordinal, 5 are discrete contiguous, and the remaining 2 are real continuous. Further details are in the preprocessing code.

All code used for the data preprocessing is in the repository `https://github.com/Epanemu/LiCE`.

# E EXPERIMENT SETUP

Here, we describe furhter details of our experiments.

## E.1 ADDITIONAL DATA CONSTRAINTS

In addition to data type constraints described in Section D, we also constrain some features for immutability and causality.

**GMSC**

- **Immutable**: `NumberOfDependents`
- **Monotone**: `age` cannot decrease
- **Causal**: no constraints

**Adult**

- **Immutable**: `race` and `sex`
- **Monotone**: `age` cannot decrease and `education` cannot decrease
- **Causal**: `education` increases $\implies$ `age` increases

**Credit**

- **Immutable**: `Number of people being liable to provide maintenance for`, `Personal status and sex`, and `foreign worker`
- **Monotone**: `Age` cannot decrease
- **Causal**: `Present residence since` increases $\implies$ `Age` increases and `Present employment since` increases $\implies$ `Age` increases

### E.2 HYPERPARAMETER SETUP

The entire configuration can be found in the code, but we also present (most of) it here.

**Neural Network** We compare methods on a neural network with four layers, first with a size equal to the length of the encoded input, then 20 and 10 for hidden layers, and a single neuron as output. It trained with batch size 64 for 50 epochs. We compare all methods on this neural network architecture, trained separately five times for each training set (from the five folds).

**SPN** To create fewer nodes in the SPN (i.e., to not overtrain it), we set the `min_instances_slice` parameter to the number of samples divided by 20.

**CE methods** We used default parameters for most methods. In cases when there were no default values set, we used the following:

- *DiCE*: we use the gradient method of searching for CEs.
- *VAE*: we set the size of the model to copy the predictor model. We parametrize the hinge loss with a margin of 0.1 and multiply the validity loss by 10 to promote validity. We use learning rate 1e-3 and batch size 64. We use weight decay of 1e-4 and train for 20 epochs (200 for the Credit data since the dataset is small).
- *FACE*: we only configure the fraction of the dataset used to search for the CE, increasing it to 0.5 for the Credit dataset due to its size.
- *C-CHVAE*: we set the size of the model to copy the predictor model. For the Credit dataset, we increase the number of training epochs to 50.
- *PROPLACE*: We create the retrained NN models to reflect the same architecture and train them for 15 epochs. We set up 1 instance of PROPLACE per class and set its delta by starting at 0.025 and decreasing by 0.005 until we are able to recover enough samples.
- *LiCE + MIO*: For our methods, we configure a time limit of 2 minutes for MIO solving. These are high enough for MIO, but constrained LiCE struggles with increasing likelihood requirements. We generate 10 closest CEs, not using the relative distance parameter. We set the decision margin $\tau = 10^{-4}$ and we use one $\epsilon_j = 10^{-4}$ for all features $j$ because they are normalized. In the SPNs, we use $T_n^{\mathrm{LL}} = 100$ as a safe upper bound though this could be computed more tightly for an individual sum node. We choose $\delta^{\mathrm{SPN}}$ equal to the median (or lower quartile) of likelihood on the dataset. For LiCE (optimize), we used $\alpha = 0.1$ since features are normalized to $[0, 1]$ and log-likelihood often takes values in the $[-100, -10]$ range.

Table 10: Comparison of LiCE variants. (optimize) means that we optimize the likelihood together with the distance, with coefficient $\alpha = 0.1$. (quartile) means that we constrain the CE to have the likelihood greater or equal to the lower quartile likelihood of training data. (median) is the same as (quartile), but we take the median instead of the quartile. Finally, (sample) is a relaxation of the (median) variant. It constrains the CE likelihood to be greater or equal to the likelihood of the factual sample or the median value, whichever is lower.

| Method | GMSC | | | Adult | | | Credit | | |
|---|---|---|---|---|---|---|---|---|---|
| | NLL | Similarity | Sparsity | NLL | Similarity | Sparsity | NLL | Similarity | Sparsity |
| MIO (+spn) | $27.9 \pm 6.6$ | $5.9 \pm 1.5$ | $2.1 \pm 0.8$ | $17.8 \pm 3.8$ | $5.8 \pm 3.8$ | $2.2 \pm 0.9$ | $43.6 \pm 17.5$ | $4.4 \pm 2.8$ | $2.3 \pm 1.1$ |
| LiCE (optimize) | $25.6 \pm 4.6$ | $5.9 \pm 1.6$ | $2.6 \pm 1.1$ | $18.1 \pm 3.9$ | $\mathbf{5.6 \pm 3.8}$ | $\mathbf{2.1 \pm 1.0}$ | $\mathbf{28.8 \pm 3.3}$ | $4.4 \pm 2.8$ | $2.3 \pm 1.2$ |
| LiCE (quartile) | $27.0 \pm 3.7$ | $\mathbf{5.8 \pm 1.5}$ | $\mathbf{1.7 \pm 0.8}$ | $18.4 \pm 3.6$ | $5.7 \pm 3.9$ | $\mathbf{2.1 \pm 1.0}$ | $39.1 \pm 15.0$ | $\mathbf{4.3 \pm 2.8}$ | $2.0 \pm 1.1$ |
| LiCE (median) | $\mathbf{18.3 \pm 2.2}$ | $11.0 \pm 3.4$ | $4.4 \pm 1.2$ | $\mathbf{12.9 \pm 1.0}$ | $9.7 \pm 6.6$ | $3.0 \pm 1.4$ | $29.9 \pm 3.1$ | $4.4 \pm 2.9$ | $2.1 \pm 1.2$ |
| LiCE (sample) | $20.5 \pm 4.2$ | $9.9 \pm 3.6$ | $4.2 \pm 1.4$ | $14.3 \pm 2.7$ | $8.5 \pm 5.8$ | $2.7 \pm 1.3$ | $31.0 \pm 6.1$ | $\mathbf{4.3 \pm 2.9}$ | $\mathbf{2.0 \pm 1.2}$ |

### E.3 COMPUTATIONAL RESOURCES

Most experiments ran on a personal laptop with 32GB of RAM and 16 CPUs AMD Ryzen 7 PRO 6850U, but since the proposed methods had undergone wider experimentation, their experiments were run on an internal cluster with assigned 32GB of RAM and 16 CPUs, some AMD EPYC 7543 and some Intel Xeon Scalable Gold 6146, based on their availability.

Regarding computational time, it is non-trivial to estimate. The time varies greatly for some methods since, for example, VAE retries generating a CE until a valid is found or a limit on tries is reached. Most methods we compared took a few hours for the 500 samples, including the method training. The MIO method takes, on average, a few seconds to generate an optimal counterfactual, while LiCE often reaches the 2-minute time limit.

Considering the tests presented in this paper, we estimate 200 hours of real-time was spent generating them, meaning approximately 3,200 CPU hours. If we include all preliminary testing, the compute time is estimated at around 20,000 CPU hours, though these are all inaccurate rough estimates, given that the hours were not tracked.

## F FURTHER COMPARISONS

In this section, we would like to discuss some results that could not fit into the article's main body.

### F.1 LiCE VARIANTS

We tested multiple versions of using the SPN within LiCE. In Table 10, we show results for 2 more configurations.

One, called *(sample)*, is a relaxation of the (median) variant. It constrains the CE likelihood to be greater or equal to the likelihood of the factual sample (i.e., the counterfactual should have, at worst, the same likelihood as the factual) or the median value, whichever is lower. This increases the proportion of factuals for which the method returns a CE in time, though only by 10 percentage points at most. This suggests that the complexity might not depend on the likelihood of the factual, thus that there might be a notable difference in likelihood landscape for the opposite classes.

The *LiCE (quartile)* is a weaker variant of *LiCE (median)*, with the bound set to the first quartile instead of the median likelihood. This is enough to obtain CEs for $100\%$ of factuals (and in good time, see Table 12). Its good performance w.r.t. similarity and sparsity is possibly caused by the method returning very close CEs with a "good enough" log-likelihood.

The results show that selecting the most likely CE out of 10 local optima given by MIO is quite strong. The two-stage setup can be quite performant. The results on similarity show that some of the MIO CEs are not globally optimal. This is because the SPN in the second phase selects some of the locally optimal (i.e., globally suboptimal) CEs.

Table 11: Results in the same format as in Table 4, but we consider only valid CEs generated for the intersection of factuals for which all methods generated a valid CE. These results are more suitable for the comparison of methods between each other. The VAE was omitted from the evaluation on the GMSC dataset because the intersection of factuals would be empty if we included VAE.

| Method | GMSC (254 factuals) | | | Adult (55 factuals) | | | Credit (56 factuals) | | |
|--------|------|------------|----------|------|------------|----------|------|------------|----------|
| | NLL | Similarity | Sparsity | NLL | Similarity | Sparsity | NLL | Similarity | Sparsity |
| DiCE (+spn) | $28.1 \pm 5.5$ | $28.5 \pm 6.5$ | $6.6 \pm 1.1$ | $19.8 \pm 2.6$ | $22.9 \pm 6.1$ | $4.1 \pm 1.4$ | $35.1 \pm 3.0$ | $22.1 \pm 4.6$ | $7.6 \pm 1.9$ |
| VAE (+spn) | - | - | - | $17.9 \pm 2.8$ | $31.9 \pm 9.7$ | $5.0 \pm 1.2$ | $46.2 \pm 17.0$ | $27.8 \pm 6.2$ | $10.7 \pm 1.8$ |
| C-CHVAE | $25.8 \pm 2.5$ | $18.2 \pm 4.8$ | $8.3 \pm 0.7$ | $17.3 \pm 3.0$ | $7.5 \pm 4.9$ | $2.6 \pm 0.8$ | $32.1 \pm 3.5$ | $12.9 \pm 4.8$ | $6.6 \pm 1.5$ |
| FACE ($\epsilon$) | $28.8 \pm 6.6$ | $15.2 \pm 4.1$ | $8.5 \pm 1.2$ | $14.1 \pm 2.6$ | $9.3 \pm 6.5$ | $2.7 \pm 1.0$ | $42.0 \pm 17.5$ | $17.7 \pm 5.0$ | $7.0 \pm 1.5$ |
| FACE (knn) | $28.5 \pm 6.9$ | $15.4 \pm 4.4$ | $8.4 \pm 1.2$ | $13.9 \pm 2.8$ | $8.9 \pm 6.0$ | $2.8 \pm 1.0$ | $42.8 \pm 17.8$ | $18.6 \pm 5.4$ | $7.1 \pm 1.5$ |
| PROPLACE | $27.8 \pm 4.5$ | $13.3 \pm 3.2$ | $6.5 \pm 1.2$ | $15.1 \pm 2.3$ | $19.2 \pm 7.0$ | $3.9 \pm 1.0$ | $37.1 \pm 14.8$ | $22.8 \pm 4.7$ | $8.8 \pm 1.2$ |
| MIO (+spn) | $27.1 \pm 6.3$ | $\mathbf{6.2 \pm 1.5}$ | $\mathbf{2.1 \pm 0.8}$ | $15.8 \pm 3.6$ | $3.1 \pm 2.2$ | $1.6 \pm 0.7$ | $34.1 \pm 12.0$ | $\mathbf{2.3 \pm 1.3}$ | $1.9 \pm 0.8$ |
| LiCE (optimize) | $24.4 \pm 5.4$ | $6.3 \pm 1.5$ | $2.6 \pm 1.1$ | $16.4 \pm 3.9$ | $\mathbf{2.9 \pm 2.2}$ | $\mathbf{1.4 \pm 0.7}$ | $\mathbf{28.8 \pm 3.4}$ | $\mathbf{2.3 \pm 1.3}$ | $1.7 \pm 0.8$ |
| LiCE (median) | $\mathbf{18.3 \pm 2.2}$ | $11.1 \pm 3.3$ | $4.3 \pm 1.2$ | $\mathbf{12.5 \pm 1.2}$ | $6.1 \pm 4.6$ | $2.1 \pm 1.1$ | $29.8 \pm 2.8$ | $\mathbf{2.3 \pm 1.3}$ | $\mathbf{1.4 \pm 0.6}$ |

## F.2 VALID CEs ON COMMON FACTUALS

Table 11 shows the results on the intersection of factuals for which all methods generated a valid CE. The proposed methods show similar differences in all metrics, as in Table 4.

Notice the comparability of DiCE results on negative log-likelihood. This suggests that the two-stage setting of generating a diverse set of CEs and then selecting the likeliest could be a viable option. On the other hand, compared to LiCE (or MIO), there is a major difference in all measures.

## F.3 TIME COMPLEXITY

Regarding the complexity of the SPN formulation, the number of variables is linearly dependent on the size of the SPN (real-valued variables). Additionally, each leaf node requires one binary variable for each bin of the histogram distribution. Sum nodes require one extra binary variable per predecessor; the total number is bounded by the number of all nodes from above, but it is typically less. The number of constraints is linearly dependent on the size of the SPN.

This is, however, difficult to translate to the algorithmic complexity of solving the MIO, which is exponential w.r.t. size of the formulation in general.

Table 12 shows the median number of seconds required to generate (or fail to generate) a CE. We see that there are stark differences between methods and also between datasets. For our methods (MIO and LiCE), we constrain the maximal optimization time to 120 seconds.

LiCE seems to be comparable on Adult as well as Credit datasets. Since MIO seems to be faster, we suggest that the main portion of the overhead is caused by solving the SPN formulation. Note that the optimizing variant of LiCE takes a long time partly to prove optimality. A (non-optimal) solution could likely be obtained even with a tighter time limit.

There also seems to be some computational overhead in constructing the formulation, which could likely be partly optimized away in the implementation.

## F.4 CE GENERATION WITH KNOWLEDGE OF THE TRUE DISTRIBUTION.

In this section, we would like to compare the CE generation methods using the true data distribution. While this distribution is generally unknown, we construct the following experiments to evaluate our method in such a scenario by forming 3 synthetic datasets.

We utilize three of the Bayesian Networks (BNs) used in Section G.4 of varying size (asia, alarm, and win95pts), choose a target variable (`dysp`, `BP`, and `Problem1`, respectively) and sample a training dataset of 10,000 samples. On this training dataset, we train an SPN and a Neural Network model, which we then utilize to generate counterfactuals for a set of 100 factuals freshly sampled from the BN. We perform this whole setup for 5 different seeds for each BN and aggregate the results.

Table 12: Median time spent on the computation of a single CE. The values above 120 in the LiCE computation are caused by computational overhead in formulating the SPN. The time limit given to the solver was 120 seconds.

| Method | GMSC | Adult | Credit |
|---|---|---|---|
| DiCE | 27.55s | 18.40s | 145.21s |
| VAE | 0.70s | 0.92s | 0.67s |
| C-CHVAE | 0.47s | 0.66s | 0.56s |
| FACE ($\epsilon$) | 9.25s | 7.25s | 5.08s |
| FACE (knn) | 6.68s | 7.12s | 5.17s |
| PROPLACE | **0.25s** | **0.35s** | **0.29s** |
| MIO | 0.80s | 1.52s | 1.56s |
| LiCE (optimize) | 132.72s | 34.39s | 3.12s |
| LiCE (quartile) | 19.27s | 10.64s | 2.71s |
| LiCE (sample) | 124.32s | 14.34s | 2.86s |
| LiCE (median) | 122.50s | 17.70s | 2.93s |

Table 13: Comparison on the asia BN. LL stands for log-likelihood. We show the mean probability directly (non-log) because of a few 0 probability counterfactuals.

| asia | LL estimate (SPN) ↑ | True probability (BN) ↑ | Similarity ↓ | Sparsity ↓ | Time [s] ↓ | % valid ↑ |
|---|---|---|---|---|---|---|
| VAE (+spn) | $-1.17 \pm 0.01$ | $2.9 \times 10^{-1} \pm 0.0$ | $3.88 \pm 3.37$ | $2.18 \pm 1.15$ | $0.02 \pm 0.00$ s | 46.6 % |
| DiCE (+spn) | $-1.17 \pm 0.01$ | $2.9 \times 10^{-1} \pm 0.0$ | $3.88 \pm 3.37$ | $2.18 \pm 1.15$ | $11.86 \pm 2.98$ s | 46.6 % |
| C-CHVAE | $-2.52 \pm 1.78$ | $1.6 \times 10^{-1} \pm 9.7 \times 10^{-2}$ | $2.37 \pm 2.37$ | $1.27 \pm 0.65$ | $0.41 \pm 0.34$ s | 46.6 % |
| FACE (knn) | $-2.42 \pm 1.47$ | $1.6 \times 10^{-1} \pm 9.7 \times 10^{-2}$ | $2.34 \pm 2.39$ | $1.25 \pm 0.60$ | $0.18 \pm 0.06$ s | 46.6 % |
| FACE ($\epsilon$) | $-5.47 \pm 2.55$ | $2.4 \times 10^{-2} \pm 4.5 \times 10^{-2}$ | $6.74 \pm 3.32$ | $2.11 \pm 0.74$ | $0.24 \pm 0.06$ s | 1.8 % |
| PROPLACE | $-3.37 \pm 1.83$ | $1.0 \times 10^{-1} \pm 9.9 \times 10^{-2}$ | $3.80 \pm 3.25$ | $1.68 \pm 0.84$ | $0.10 \pm 0.03$ s | 41.8 % |
| MIO (+spn) | $-1.52 \pm 0.85$ | $2.3 \times 10^{-1} \pm 6.5 \times 10^{-2}$ | $2.89 \pm 1.99$ | $1.86 \pm 0.76$ | $0.96 \pm 0.49$ s | 100 % |
| LiCE (med) | $-1.33 \pm 0.15$ | $2.4 \times 10^{-1} \pm 4.5 \times 10^{-2}$ | $3.09 \pm 2.51$ | $1.94 \pm 0.91$ | $0.71 \pm 0.10$ s | 100 % |
| LiCE ($\alpha = 1$) | $-1.89 \pm 0.93$ | $1.8 \times 10^{-1} \pm 7.0 \times 10^{-2}$ | $2.24 \pm 2.06$ | $1.41 \pm 0.76$ | $1.00 \pm 0.32$ s | 100 % |

In the tables below (Tables 13, 14, and 15 for asia, alarm, and win95pts, respectively), we evaluate the mean log-likelihood of generated CEs using the fitted SPN, the mean true probability, mean distance (similarity), sparsity, and time spent generating the valid counterfactuals. Finally, we show the percentage of factuals for which a valid counterfactual was found by a given method.

We see that LiCE methods, especially the likelihood-optimizing variant ($\alpha = 1$), perform comparably to other methods even when taking into account the true distribution.

Finally, note that:

- performance improvements in terms of distance and sparsity reflect experiments on real data;
- only MIO-based methods generate 100% of valid counterfactuals, other methods generate 55.8% CEs at best;
- the time complexity of LiCE is on par with other methods;
- while not perfect, SPN likelihood generally correlates with the true probability (see the discussion in Section G.4 for additional details).

**Statistical significance**  To evaluate the statistical significance of our results, we rank the methods using the true probability of the generated CE. To account for cases where no CE was generated, we rank the methods that did not return a valid CE as last. We evaluate each simulated dataset (from each BN) separately. Friedman test rejects the null hypothesis that all methods perform the same with $p < 0.001$.

The average ranks are shown in Table 16 and in Figure 2, we show plots inspired by (Demšar, 2006, fig. 1a), where average ranks and results of the Nemenyi test are shown.

Table 14: Comparison on the alarm BN. LL stands for log-likelihood. We show the mean probability directly (non-log) because of a few 0 probability counterfactuals. DiCE did not return any valid counterfactuals.

| alarm | LL estimate (SPN) ↑ | True probability (BN) ↑ | Similarity ↓ | Sparsity ↓ | Time [s] ↓ | % valid ↑ |
|---|---|---|---|---|---|---|
| VAE (+spn) | $-18.32 \pm 4.49$ | $1.1 \times 10^{-7} \pm 2.0 \times 10^{-7}$ | $27.15 \pm 11.31$ | $8.27 \pm 2.96$ | $0.02 \pm 0.00$ s | 53.2 % |
| DiCE (+spn) | - | - | - | - | - | 0 % |
| C-CHVAE | $-9.72 \pm 4.31$ | $3.0 \times 10^{-3} \pm 5.6 \times 10^{-3}$ | $12.61 \pm 8.12$ | $3.68 \pm 2.11$ | $0.72 \pm 0.59$ s | 27 % |
| FACE (knn) | $-8.59 \pm 3.57$ | $2.9 \times 10^{-3} \pm 5.0 \times 10^{-3}$ | $12.25 \pm 8.34$ | $3.53 \pm 2.13$ | $7.69 \pm 1.92$ s | 54 % |
| FACE ($\epsilon$) | $-9.28 \pm 3.40$ | $1.7 \times 10^{-3} \pm 4.0 \times 10^{-3}$ | $13.61 \pm 8.18$ | $3.92 \pm 2.08$ | $7.29 \pm 1.87$ s | 40.2 % |
| PROPLACE | $-10.38 \pm 3.99$ | $1.0 \times 10^{-3} \pm 2.8 \times 10^{-3}$ | $14.79 \pm 9.83$ | $4.29 \pm 2.54$ | $0.56 \pm 0.14$ s | 54 % |
| MIO (+spn) | $-10.92 \pm 5.29$ | $2.9 \times 10^{-3} \pm 5.6 \times 10^{-3}$ | $3.86 \pm 1.37$ | $1.37 \pm 0.53$ | $1.97 \pm 0.18$ s | 100 % |
| LiCE (med) | $-7.72 \pm 1.87$ | $2.8 \times 10^{-3} \pm 5.3 \times 10^{-3}$ | $8.42 \pm 7.41$ | $2.52 \pm 1.98$ | $10.68 \pm 13.95$ s | 100 % |
| LiCE ($\alpha = 1$) | $-9.49 \pm 4.57$ | $3.3 \times 10^{-3} \pm 5.8 \times 10^{-3}$ | $4.58 \pm 2.52$ | $1.51 \pm 0.76$ | $8.67 \pm 2.31$ s | 100 % |

Table 15: Comparison on the win95pts BN. LL stands for log-likelihood. We show the mean probability directly (non-log) because of a few 0 probability counterfactuals. VAE did not return any valid counterfactuals, and DiCE returned the same counterfactuals for all factuals, leading to very poor sparsity and similarity.

| win95pts | LL estimate (SPN) ↑ | True probability (BN) ↑ | Similarity ↓ | Sparsity ↓ | Time [s] ↓ | % valid ↑ |
|---|---|---|---|---|---|---|
| VAE (+spn) | - | - | - | - | - | 0 % |
| DiCE (+spn) | $-151.40 \pm 4.65$ | $0.0 \pm 0.0$ | $1.8 \times 10^{5} \pm 3.8 \times 10^{5}$ | $63.50 \pm 3.99$ | $39.17 \pm 10.84$ s | 44.2 % |
| C-CHVAE | $-7.38 \pm 2.73$ | $3.0 \times 10^{-4} \pm 4.7 \times 10^{-4}$ | $7.89 \pm 7.26$ | $3.87 \pm 2.65$ | $1.08 \pm 0.55$ s | 55.8 % |
| FACE (knn) | $-8.59 \pm 3.19$ | $1.5 \times 10^{-3} \pm 2.4 \times 10^{-3}$ | $6.98 \pm 5.33$ | $3.44 \pm 1.88$ | $8.16 \pm 2.54$ s | 55.8 % |
| FACE ($\epsilon$) | $-10.30 \pm 3.65$ | $7.2 \times 10^{-4} \pm 1.6 \times 10^{-3}$ | $10.05 \pm 5.96$ | $4.51 \pm 1.97$ | $7.23 \pm 1.46$ s | 28.8 % |
| PROPLACE | $-8.20 \pm 2.76$ | $3.2 \times 10^{-4} \pm 5.0 \times 10^{-4}$ | $7.08 \pm 7.05$ | $3.56 \pm 2.80$ | $0.49 \pm 0.15$ s | 55.8 % |
| MIO (+spn) | $-10.86 \pm 4.54$ | $9.1 \times 10^{-5} \pm 1.9 \times 10^{-4}$ | $2.43 \pm 0.71$ | $1.70 \pm 0.46$ | $1.85 \pm 0.17$ s | 100 % |
| LiCE (med) | $-7.77 \pm 1.22$ | $5.2 \times 10^{-4} \pm 2.1 \times 10^{-3}$ | $7.42 \pm 8.58$ | $3.47 \pm 3.15$ | $5.26 \pm 0.47$ s | 100 % |
| LiCE ($\alpha = 1$) | $-9.93 \pm 4.15$ | $1.3 \times 10^{-3} \pm 6.0 \times 10^{-3}$ | $2.49 \pm 1.72$ | $1.57 \pm 0.83$ | $5.48 \pm 0.58$ s | 100 % |

The thresholding variant of LiCE ranks the highest on all simulated datasets, and the Nemenyi test with confidence level $\alpha = 0.05$ cannot reject its similarity only from MIO (+spn) on the data sampled from asia and win95pts.

To be more generous towards competing methods, we could consider only factuals for which both methods successfully returned a valid CE. This disadvantages LiCE variants and MIO (+SPN) because they are the only methods that always succeed in generating a valid CE. The Friedman test also rejects the null hypothesis with $p < 0.001$. The results of Nemenyi test are shown in Figure 3, in a similar setup to Figure 2.

The thresholding variant of LiCE still achieves the highest rank for alarm and asia BNs. However, its performance is not statistically better than other CE methods. On win95pts, our methods rank poorly, a striking contrast to Figure 2. The plausibility of CEs returned by the proposed methods clearly depends on the quality of the SPN. It is possible that for such a big BN, generating 10,000 points with one of $2^{76}$ possible values is not enough to have a high-quality SPN using the LearnSPN algorithm. Some methods are omitted from Figure 3 due to a low number of factuals in the intersection.

## G  FURTHER COMMENTS

Given the limited size of the Credit dataset, it is unsurprising to see so many failures of some methods. There is not much data for some methods to support the training. This might be behind the low success rate of computing a valid CE.

Table 16: Average ranks of CE methods for each simulated dataset. We rank the methods based on the true probability, evaluated by the BN.

| | VAE (+spn) | DiCE (+spn) | C-CHVAE | FACE (knn) | FACE ($\epsilon$) | PROPLACE | MIO (+spn) | LiCE (med) | LiCE ($\alpha = 1$) |
|---|---|---|---|---|---|---|---|---|---|
| asia | 4.866 | 4.866 | 6.242 | 6.221 | 7.603 | 6.672 | 2.529 | 2.350 | 3.651 |
| alarm | 6.789 | 7.642 | 6.263 | 4.672 | 5.369 | 5.427 | 3.570 | 2.254 | 3.014 |
| win95pts | 7.702 | 6.029 | 5.146 | 4.341 | 6.054 | 4.916 | 3.717 | 3.209 | 3.886 |

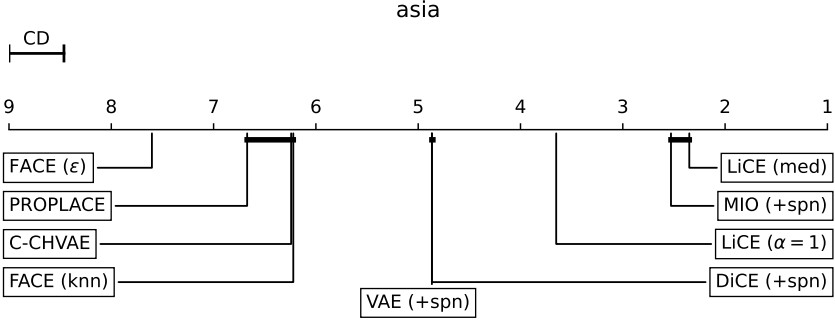

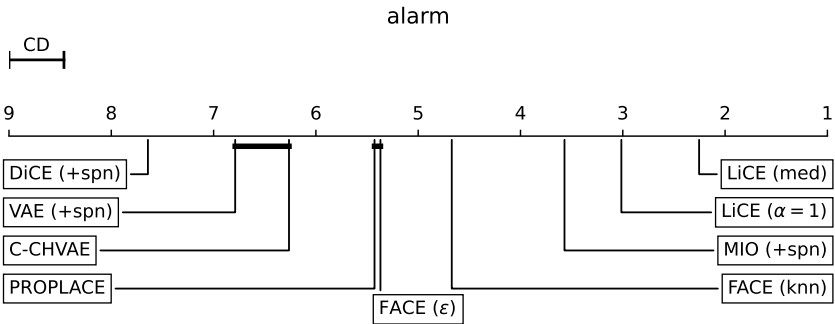

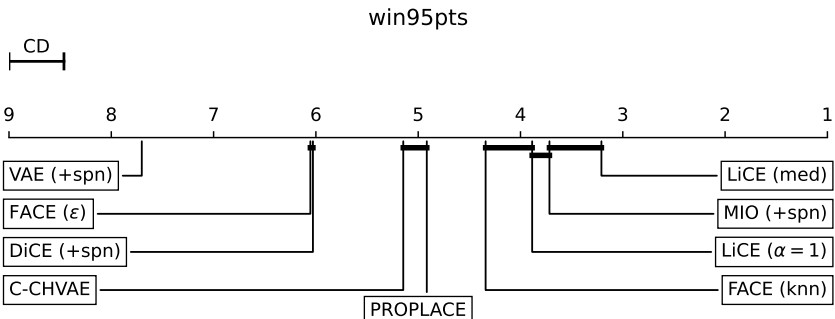

Figure 2: Average ranks and results of Nemenyi test for the true probability of CEs generated for factuals sampled using the Bayesian Networks. When a method fails to generate a valid CE, we give it the lowest rank. Groups of methods that are not significantly different (using the Nemenyi test with $\alpha = 0.05$) are connected. Critical difference (for $\alpha = 0.05$) is shown on the upper left.

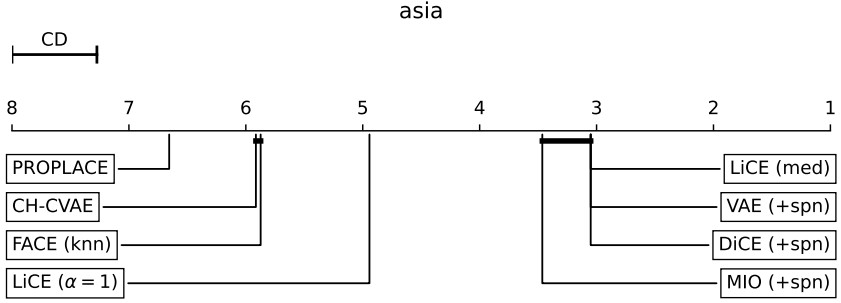

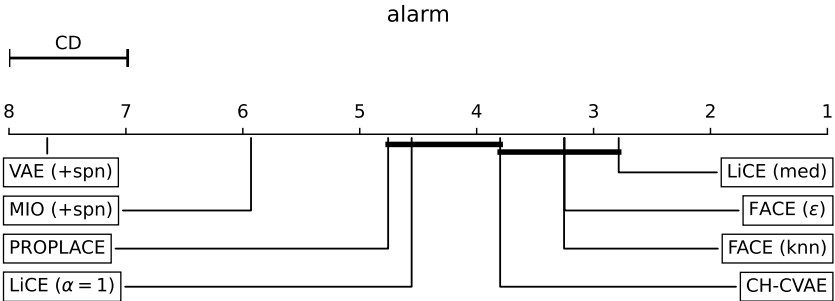

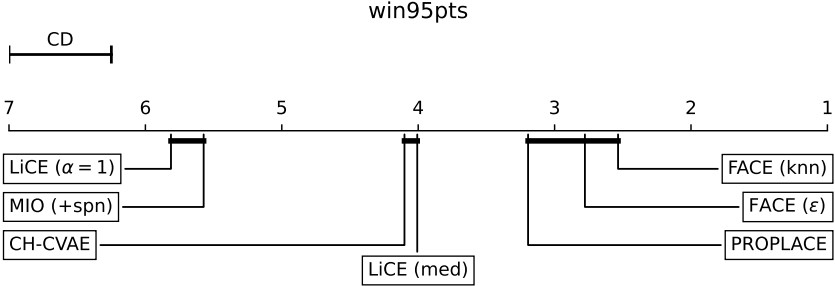

Figure 3: Average ranks and results of Nemenyi test for the true probability of CEs generated for factuals sampled using the Bayesian Networks. Here, we consider only factuals where each CE method was successful. FACE ($\epsilon$) was removed from comparison on asia, because it returned a CE only for 6 factuals. DiCE was removed from alarm because it did not return any CEs, and from win95pts because there were no factuals in the intersection of successful CEs. VAE was removed from win95pts because it did not return any CEs. Groups of methods that are not significantly different (using the Nemenyi test with $\alpha = 0.05$) are connected. Critical difference (for $\alpha = 0.05$) is shown on the upper left.

Regarding the other results, it is possible that the VAE method has been misconfigured for GMSC, returning very few results.

The main disadvantage of LiCE is the time complexity of CE generation. We argue, however, that for some use cases, the user might be willing to wait to obtain a high-quality CE. We leave this decision to the user.

## G.1 OMITTED METHODS

It is not feasible to test against all CE methods, so we looked for a selection of methods that consider the plausibility of generated CEs. Two methods were, however, not tested for the following reasons:

- PlaCE (Artelt & Hammer, 2020) does not allow for explaining Neural Networks. It also cannot model categorical features well.
- DACE (Kanamori et al., 2020) does not have a public implementation that would allow for Neural Networks as models. It might also struggle with the size of datasets used here since they are an order of magnitude larger, and DACE computes the Local Outlier Factor, meaning that the formulation size increases linearly with the increase in the number of samples.

## G.2 POTENTIAL NEGATIVE CONSEQUENCES

Given the many CE methods for generating CEs, one must deal with the disagreement problem (Brughmans et al., 2024), where a user could be misled by the owner of an ML model who selects the CEs that align with their interests. We argue that our method does not severely contribute to this problem, since it is deterministic, thus resistant to re-generation attempts to obtain a more favorable CE. Our method also outperforms many other methods, making arguing for their use more difficult.

## G.3 SUITABILITY OF SPNS FOR ESTIMATING THE PLAUSIBILITY OF CES

We believe that SPNs are well suited for the problem because (i) they naturally model distributions over continuous and discrete random variables; (ii) their simple formulation can be tightly approximated within MIO; (iii) and they are universal approximators (Nguyen & McLachlan, 2019).

Other options are

- *Gaussian Mixture Models (GMMs)* are designed only for continuous random variables. SPNs are a strict superset to GMMs.
- *Flow models* are very flexible, but they model only distribution on continuous random variables. Since they are parametrized by neural networks, they might be rather difficult to formulate within MIO, especially considering their block nature relying on smooth non-linear functions (exp, softplus).
- *Neural auto-regressive models* can model discrete and random variables, and they provide exact likelihood. But again, they use relatively large neural networks, which might need non-linearities that are difficult to use within MIO (sigmoid, softmax).
- *Auto-encoders* have, with respect to MIO similar advantages and disadvantages as neural auto-regressive models. Furthermore, they provide only lower-bound estimates of true likelihood in the form of ELBO.

## G.4 SPN AS A MODEL OF DATA DISTRIBUTION

Furthermore, we test the ability of an SPN to model the true data distribution empirically. We choose 8 Bayesian Networks (BNs) to model the data-generating process. For each of them, we generate (sample) training data, then fit an SPN to this data, and finally compare the SPN's likelihood estimates of test samples to their true probability, given by the BN.

More specifically, we utilize the `bnlearn` Python library (Taskesen, 2020) and select 7 Bayesian Networks of varying sizes from the Bayesian Network Repository (Scutari, 2010) (namely asia,

Table 17: Size comparison of BNs used to generate the synthetic data.

|  | sprinkler | asia | sachs | child | water | alarm | win95pts | andes |
|---|---|---|---|---|---|---|---|---|
| Number of nodes | 4 | 8 | 11 | 20 | 32 | 37 | 76 | 223 |
| Number of edges | 4 | 8 | 17 | 25 | 66 | 46 | 112 | 338 |
| Number of parameters | 9 | 18 | 178 | 230 | 10083 | 509 | 574 | 1157 |

Table 18: Evaluation of the fit by correlation coefficients for all 8 tested BNs. Total Variation was computed only for smaller BNs, where the computation was practical. Numbers are rounded to 3 decimal digits.

|  | sprinkler | asia | sachs | child | water | alarm | win95pts | andes |
|---|---|---|---|---|---|---|---|---|
| Pearson coefficient | 0.996 | 0.990 | 0.964 | 0.954 | 0.955 | 0.959 | 0.959 | 0.793 |
| Kendall ($\tau$-b) coeff. | 1.000 | 0.992 | 0.860 | 0.853 | 0.828 | 0.892 | 0.891 | 0.600 |
| Spearman coefficient | 1.000 | 1.000 | 0.972 | 0.966 | 0.961 | 0.978 | 0.977 | 0.788 |
| Total variation | 0.017 | 0.073 | 0.260 | - | - | - | - | - |

sachs, child, water, alarm, win95pts, and andes). The eighth BN (sprinkler) is another standard BN, available directly in the `bnlearn` library. See Table 17 for parameters of the used networks.

To train the SPN, we sample 10,000 points using the BN and train the SPN in the same way as for LiCE, with default parameters. Then we sample 1,000 more samples and evaluate their likelihood using the trained SPN. We also compute their true probability from the BN and compare these pairs of values. We perform the above process 5 times with different seeds for each BN and select the best-performing SPN for comparison.

In Table 18, we show the correlation coefficients of the log-likelihood and log-probability computed on the 1,000 test samples. We also show the total variation for the smaller BNs, when the value can be computed in reasonable time. All correlation evaluations reject the null hypothesis that there is no or negative correlation with $p < 0.001$. In Figure 4, we show scatter plots of the data on which the correlation coefficients were computed. The BNs are sorted in increasing order of the number of nodes.

The SPN performs quite well, with the exception of the biggest BN (andes), where the drop might be explained by 10,000 samples (from $2^{223}$ possible values) being too few to train the SPN precisely.

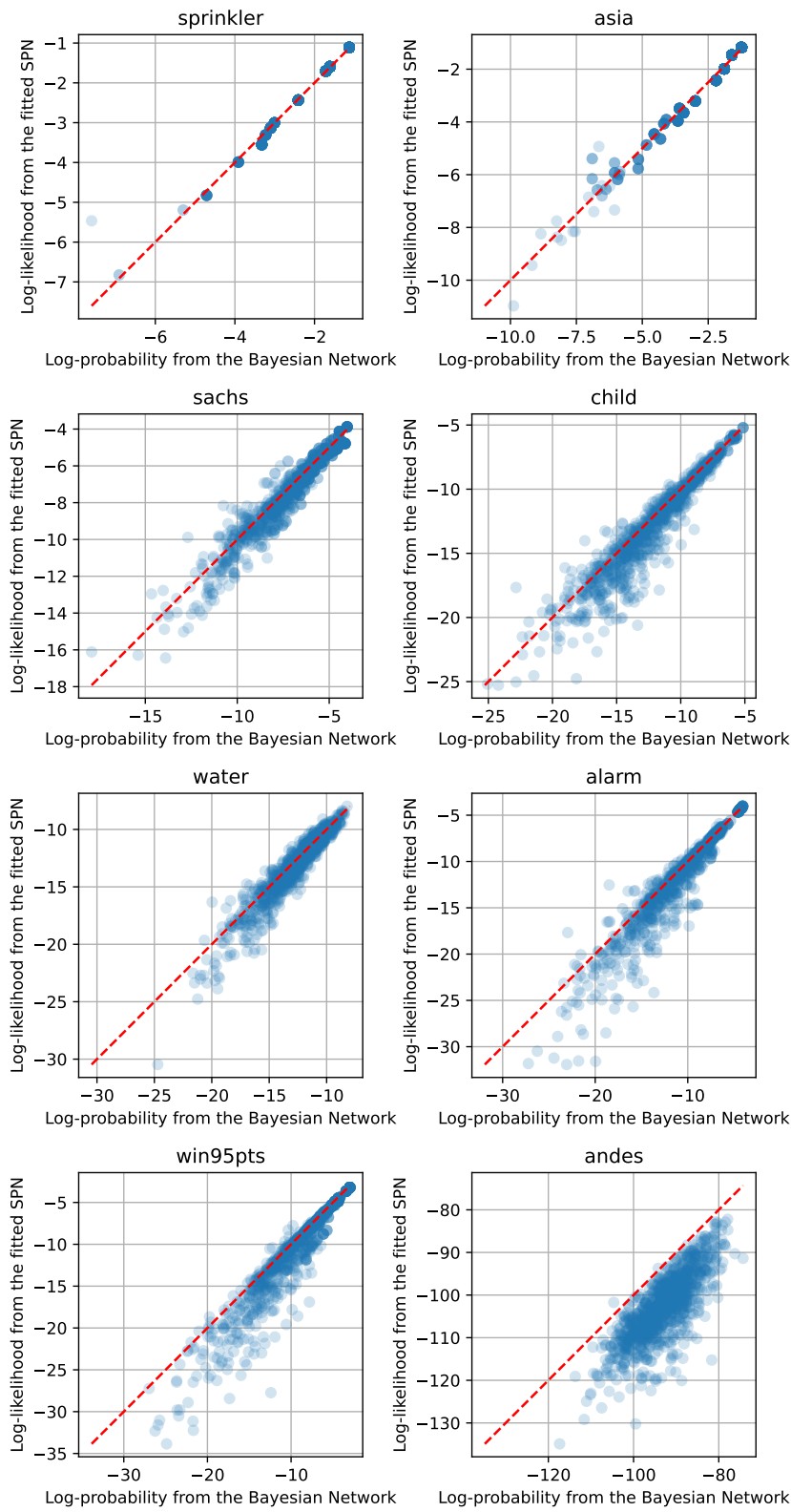

Figure 4: Visual comparison of correlation between the true probability of a sample and log-likelihood estimate given by an SPN. Each plot shows 1,000 points sampled using the given Bayesian Network, see their names in the titles. For numerical comparison, see Table 18.

