# OpenReview forum: "Generating Likely Counterfactuals Using Sum-Product Networks"
_ICLR.cc/2025/Conference — ICLR 2025 Poster_

### Official Review · Reviewer_WAgz · 2024-11-01

**Soundness:** 3
**Presentation:** 3
**Contribution:** 3
**Rating:** 8
**Confidence:** 5

**Summary:**

The paper proposed a method for computing plausible counterfactual explanations by using sum-product networks.

**Strengths:**

The paper addresses an important problem and proposes a novel and interesting approach. Its strength is its ability to handle categorical features (often encountered in real-world data sets).

**Weaknesses:**

Disclaimer: I have reviewed the paper before (Neurips 2024)—I hope that this does not disqualify me as a reviewer. In my opinion, the paper has improved significantly. In particular, the empirical evaluation is now done fairly, the notation and formalization are clear now, and the structure of the entire paper is now more accessible and more "scientific" compared to the previous version that I reviewed.

However, I still think that some more information on SPNs could be added: How powerful are SPNs? What types of distribution can they model? What are the assumptions? How to estimate SPN's parameters from data? Given the large number of existing methods (for computing plausible counterfactuals), this information would help practitioners select one of those existing methods. There is information given in the appendix but it might be also good to (briefly) answer some of those questions in the main text.

I understand that there is not enough space to fully introduce SPNs but I still think that it would be important to highlight what they can model and what they can not -- this would help practitioners to select an appropriate method for their particular use case. Currently, different methods exist for computing plausible counterfactuals based on Gaussian Mixture Models, Kernel Density Estimators, Neural Autoencoders, etc., and each of those methods has its advantages and disadvantages. However, I am happy to discuss this and step back if the authors or the other reviewers can convince me that this is not necessary or is somehow already in the paper. Thanks :)

**Questions:**

See weaknesses.

---

> ### Author Response · Authors · 2024-11-19
> **Rebuttal**
>
> We thank the reviewer for their review and are pleased that they find the paper significantly improved.
>
> We address their questions bellow:
>
> > How powerful are SPNs? What types of distribution can they model? What are the assumptions?
>
> Since this question is similar to one of the questions of Reviewer RaSc, we copy the answer below:
>
> We believe that SPNs are well suited for the problems, because (i) they naturally model distribution over continuous and discrete random variables; (ii) their simple formulation can be tightly approximated within MIO; (iii) and they are universal approximators (Nguyen & McLachlan, 2018).
>
> Other options are
>  - _Gaussian Mixture Models (GMMs)_ are designed only for continuous random variables. SPNs are a strict superset to GMMs.
>  - _Flow models_ are very flexible, but they model only distribution on continuous random variables. Since they are parametrized by neural networks, they might be rather difficult to formulate within MIO, especially considering their block nature relying on smooth non-linear functions (exp, softplus).
>  - _Neural auto-regressive models_ can model discrete and random variables, and they provide exact likelihood. But again, they use relatively large neural networks which might need non-linearities that are difficult to use within MIO (sigmoid, softmax).
>  - _Auto-encoders_ have, with respect to MIO, similar advantages and disadvantages as neural auto-regressive models. Furthermore, they provide only lower-bound estimates of true likelihood in the form of ELBO.
>
> We state the main advantages of SPNs on lines 82 (in Introduction) and 251 (in Related work). We agree that more details regarding this question are needed. Thus, we have added the above discussion to Appendix A.8.3.
>
> We are not aware of any thorough comparisons of SPNs to other models. Here, we list a few publications providing a comparison to some popular models.
>  - Comparison to Bayesian Networks and Markov models (Gens & Domingos, 2013) https://proceedings.mlr.press/v28/gens13.pdf
>  - The paper introducing SPTN where even the vanilla SPN is favorably compared to Masked auto-regressive flow models and GMMs. https://proceedings.mlr.press/v138/pevny20a.html
>  - Comparison to Hybrid Bayesian Networks (for mixed data)  https://cdn.aaai.org/ojs/11731/11731-13-15259-1-2-20201228.pdf
>  - Comparison to VAE https://ojs.aaai.org/index.php/AAAI/article/view/25883
>  - Comparison to Masked autoencoders (MADE) and VAE https://proceedings.mlr.press/v115/peharz20a/peharz20a.pdf
>  - Related Poisson-SPNs compared to Poisson Dependency Networks and Latent Dirichlet Allocation https://ojs.aaai.org/index.php/AAAI/article/view/10844
>
> Nguyen, H. D., & McLachlan, G. (2018). On approximations via convolution-defined mixture models. Communications in Statistics - Theory and Methods, 48(16), 3945–3955. https://doi.org/10.1080/03610926.2018.1487069
>
> > How to estimate SPN's parameters from data?
>
> Estimating SPNs from data is an active and fruitful area of research. See the survey by Xia et al. (2023). Our method is independent of the algorithm used to learn the SPN, and we utilize a widely used implementation of LearnSPN.
>
> Additionally, to empirically validate the quality of an SPN as a model of data likelihood, one would need to perform experiments on synthetic data from known underlying distributions. We agree that this would strengthen the paper and we are working on such experiments. If there is some distribution you suggest we would welcome it.

---

> > ### Comment · Reviewer_WAgz · 2024-11-20
> >
> > Thanks for your response.
> > Yeah, as I said -- just put a short statement there and I am happy. Will increase my score -- fingers crossed.

---

> > > ### Author Response · Authors · 2024-11-20
> > >
> > > We have added the summary, as written in our rebuttal above, to Appendix A.8.3. We have also added a reference to it to the main body at the end of the Related Work section (line 253). Do you find that sufficient, or would you prefer us to expand on it?

---

> > > > ### Comment · Reviewer_WAgz · 2024-11-20
> > > >
> > > > It's sufficient. Thanks!

---

### Official Review · Reviewer_kD9A · 2024-11-03

**Soundness:** 3
**Presentation:** 2
**Contribution:** 2
**Rating:** 5
**Confidence:** 3

**Summary:**

This paper studies the problem of generating counterfactual explanations for a fixed classification function, say $f(x)$. The explanation here refers to different input $x'$ to the classifier $f(x)$ (with a factual baseline $x$) such that they result in different classification outcomes $f(x') \neq f(x)$. In general, there are many such realizations of input $x'$. This paper studies the problem of finding the most likely counterfactual explanations $x'$ satisfying a set of selection desiderata, including validity, similarity, sparsity, and actionability. The authors evaluate the likelihood of an input $x'$ by fitting a sum-product network (SPN). Using the formalism of SPNs, the selection desiderata could be translated into a series of equivalent polynomial constraints. The explanation generation is then reduced to an equivalent mixed integer optimization (MIO) problem.

**Strengths:**

- The writing is generally clear. The paper is well-organized. While no main theorem is presented in the paper, the derivation of the mixed integer program is technically sound.
- Comprehensive simulations were performed. Simulation results support the proposed MIO formulation.
- The authors have clearly stated the limitations of the proposed methods, including the computational challenges of solving MIO programs.

**Weaknesses:**

- The SPN models might be limited. First, it can only account for discrete features, while most prediction tasks in practice involve high-dimensional, continuous feature inputs. Also, SPNs require structural constraints. It could be challenging to learn SPNs that best fit with the observed data.
- The explanation generation problem is reduced to an MIO program, which is generally NP-hard to solve. This presents challenges in generalizing the proposed methods to more complex domains.
- The reduction of SPN to MIO programs is not surprising. It has been known that constrained optimization in Bayesian networks is generally equivalent to linear integer programming. This paper would be most improved by proposing an efficient approximation algorithm to solve the MIO program while leveraging the graphical structure of the learned SPN.

**Questions:**

1. Could the proposed method apply to other generative models measuring the likelihood, e.g., Gaussian processes?
2. For all variants of the proposed methods, LiCE (optimize) seems to perform the best. However, its implementation is somewhat ambiguous. The authors stated, "we optimize a combination of distance and likelihood with α = 0.1 and relax the plausibility constraint (Eq. 13)." Could the authors further elaborate on this statement? For instance, what is the "combination of distance" and how to "relax the plausibility constraint?" A pseudo-code description would be appreciated.

---

> ### Author Response · Authors · 2024-11-19
> **Rebuttal**
>
> We thank the Reviewer kD9A for their review and address their questions below:
>
> > The SPN models might be limited.
>
> Mixture models are known to be universal approximators (Nguyen & McLachlan, 2018). Since SPNs are a strict superset of Mixture models, they inherit the same property.
>
> Nguyen, H. D., & McLachlan, G. (2018). On approximations via convolution-defined mixture models. Communications in Statistics - Theory and Methods, 48(16), 3945–3955. https://doi.org/10.1080/03610926.2018.1487069
>
> > First, it can only account for discrete features [...]
>
> SPNs can model distributions over both continuous and discrete random variables and their combinations. However, it is true that our MIO formulation requires distributions over discrete variables. We can approximate any continuous random variable with a discrete random variable via discretization. Because we discretize the continuous random variable after the fitting of the SPN, we control the error in the discretization.
>
> > It could be challenging to learn SPNs that best fit with the observed data.
>
> We would like to argue that:
>  - Despite the structural constraints, SPNs are a universal approximator for probability distributions (absolutely continuous measures) (Nguyen & McLachlan, 2018).
>  - There has been much recent progress in finding the best SPN to model the data (“fitting SPN”) and learning the SPN structure is a thoroughly studied topic. See, e.g., the review by Xia et al. (2023), or the 300+ citations on the LearnSPN (Gens & Domingos, 2013).
> Out of the recent work, we would like to highlight the paper by Correia, A. H. et al. (2023).
>  - We have utilized LearnSPN, the most widely known algorithm, to showcase the performance of our formulation in a familiar setting. The actual choice of the library to utilize is orthogonal to our formulation.
>
> Correia, A. H., Gala, G., Quaeghebeur, E., de Campos, C., & Peharz, R. (2023). Continuous Mixtures of Tractable Probabilistic Models. Proceedings of the AAAI Conference on Artificial Intelligence, 37(6), 7244-7252. https://doi.org/10.1609/aaai.v37i6.25883
>
> > [...] is generally NP-hard to solve. [...]
>
> Despite the problem being NP-hard, we can generate high-quality CEs in a reasonable time. MIO solver capabilities are also improving dramatically and problems that were previously “unsolvable” can now be solved in seconds (Koch et al., 2022). Furthermore, we do not require the formulation to be solved to optimality to obtain high-quality CEs (see A.7.3).
>
> We would like to point out that MIO has been used for CE generation since the initial work in the field (Cui et al., 2015) and is being used even in the current more mature stages (Russell, 2019; Kanamori et al., 2020; Parmentier & Vidal, 2021; Mohammadi et al., 2021; Jiang et al., 2024).
>
> Koch, T., Berthold, T., Pedersen, J., & Vanaret, C. (2022). Progress in mathematical programming solvers from 2001 to 2020. EURO Journal on Computational Optimization, 10, 100031.
>
> > The reduction of SPN to MIO programs is not surprising. [...]  This paper would be most improved by proposing an efficient approximation algorithm [...]
>
> While the MIO formulation of an SPN might not be surprising to some, it has not yet been proposed in any of the published papers we found. We would also argue that the underperformance in sparsity and similarity of current methods for plausible CEs had elements of surprise to us.
> We agree that the study of potential approximation algorithms is highly relevant and we have added it to future work in the Conclusion.
>
> > Could the proposed method apply to other generative models measuring the likelihood, e.g., Gaussian processes?
>
> We believe that this is an important consideration for further work. A pioneering study by Trapp et al. (2018) shows a possible approach. We have added this to the discussion in the Conclusion.
>
> Martin Trapp, Robert Peharz, Carl E Rasmussen, and Franz Pernkopf. Learning Deep Mixtures of Gaussian Process Experts Using Sum-Product Networks. arXiv preprint arXiv:1809.04400, 2018. https://arxiv.org/abs/1809.04400
>
> > [...] what is the "combination of distance" and how to "relax the plausibility constraint?" A pseudo-code description would be appreciated.
>
> Relaxing a constraint in this context refers to removing the constraint from the model. Setting $\alpha = 0.1$ refers to the parameter in the objective function (Eq. 14) described in the paragraph **Full LiCE model**. Combination refers to the linear combination of the distance and the log-likelihood estimate in the objective function (Eq. 14).
>
> Since these are modifications of the MIO formulation, we argue that pseudo-code is not a practical description tool.

---

> > ### Author Response · Authors · 2024-11-25
> > **Ping**
> >
> > Dear kD9A --
> >
> > We would be genuinely keen on discussing the matters further.
> >
> > More broadly, we would like to point out that:
> >
> > while the problem is, indeed, NP-Hard, mixed-integer programming (MIP) is the best-performing and best-understood approach to solving a wide variety of NP-Hard optimization problems. Famously, the travelling salesman problem has been solved for instances on tens of thousands of cities to optimality using Concorde:
> > https://www.math.uwaterloo.ca/tsp/concorde/benchmarks/bench99.html
> > https://press.princeton.edu/books/hardcover/9780691129938/the-traveling-salesman-problem
> > where both variables and inequalities are added dynamically to MIP. Similarly, in bioinformatics, operations research, and transportation research, huge instances are solved on a day-to-day basis.
> >
> > The question of whether the formulation of SPN-based approximation of likelihood as a MIP is surprising or not is a moot point. Our argument is that the formulation is original and practically relevant, as our experimental results show. There certainly are other papers utilizing MIP in probabilistic models, e.g., learning Bayesian networks (https://www.sciencedirect.com/science/article/pii/S0888613X13002089), or learning dynamic Bayesian networks (https://arxiv.org/abs/2410.16100), but we stress that the  SPN-based approximation of likelihood via MIP has not been proposed previously, as far as we know, and thus is original.
> >
> > Formulating SPN-based approximation of likelihood as a MIP also makes it possible to use this approach to optimization under uncertainty across a number of other applications. While rather far removed from the focus of the present submission, this could also be the most important "take away" message for audiences beyond explainable AI.

---

> > ### Comment · Reviewer_kD9A · 2024-12-02
> >
> > I have read through the authors' responses and other reviewers' comments. The authors have addressed my concerns about the expressive power of SPN and learning SPN from observed data. However, my concerns about the computational complexity of finding plausible counterfactual explanations remain. Due to these reasons, I am raising my score to 5.

---

### Official Review · Reviewer_RaSc · 2024-11-03

**Soundness:** 2
**Presentation:** 2
**Contribution:** 2
**Rating:** 5
**Confidence:** 3

**Summary:**

The paper presents a novel method to model counterfactual explanation (CE) search that maximizes likelihood, closeness, and sparseness. The method leverages sum-product networks (SPNs) to estimate point likelihoods (or plausibility) and use mixed integer optimization (MIO) to optimize the different objectives simultaneously. The work includes empirical comparisons with state-of-the-art CE algorithms.

**Strengths:**

- The use of SPNs to model plausibility in counterfactual explanations is a novel and creative approach, addressing limitations in handling mixed data that many existing methods struggle with.

- The integration of SPNs and MIO to optimize multiple constraints simultaneously offers flexibility and the formulation of bounds for the output of SPNs that can be integrated as constraints into a MIO formulation is (as the authors mention) quite interesting on its own.

**Weaknesses:**

- The key concept of plausibility, derived from the output of SPNs, is assumed to be correct without evaluation. This is a critical assumption, as it underpins the entire method, and its validity should be examined and justified. The lack of empirical evaluation of this concept weakens the overall contribution.

- The experiments appear to be biased in their evaluation of plausibility as the SPN distribution is taken as the true one, giving the proposed method an unfair advantage over the baselines

- The clarity of the paper could be improved, particularly around the explanation of key terms and concepts like plausibility and outliers, which are not sufficiently defined when first introduced.

- The method appears computationally expensive, which could be a practical concern in real-world applications. This issue is not directly addressed in the paper, but the experiments suggest that the optimization process may be time-consuming, particularly for more complex scenarios. The paper would benefit from a discussion of the computational cost and potential ways to mitigate it, such as pruning strategies, approximations, or discussing scalability.

**Questions:**

Content-related:

- L53: What is the definition of an outlier in your context? Are you suggesting that an epsilon-ball around the factual could contain outliers due to factor interactions and their joint distribution? Please expand on this.

- L80: Table 1 is not explained. Why are the "other desiderata" important?

- L97-107: A definition of plausibility is missing in this section. You claim that DiCE is the most plausible—what criteria support this claim? Additionally, LiCE changes installment/disposable income while keeping amount and duration constant, suggesting a change in income. Is this more "realizable" than borrowing less? MIO changes two features (duration and income); could you provide more details on how these changes are measured?

- Later in the paper, it becomes clearer that your definition of plausibility stems from the joint probability distribution extracted from SPNs. However, this raises concerns about whether this plausibility measure should be evaluated independently. In your experiments, you assume that this probability distribution provides the correct measure of plausibility without empirical validation. Could you justify this assumption further, or better yet, provide an evaluation of plausibility as a standalone concept? This is crucial, as your entire method relies on the assumption that SPN-based plausibility is the most accurate or appropriate for counterfactuals.

- L184: Why do you take the median for categorical values? Could you use their actual values instead? Taking the median for categorical variables may cause all of them to start from the same point.

- L188: Why will at least one always be 0? Please provide intuition. Also, the citation is placed after the full stop—should be corrected.

- L237: Outliers are mentioned again without a definition. Also, plausibility is referenced but not defined earlier. An intuition would help the reader follow the motivation more clearly.

- L260: Why do you work in log-space? Is it to simplify handling products? Please provide a rationale for this choice.

- L332: Taking a threshold of 0 on the raw output of a neural network implies a threshold probability of 0.5 after applying a logit. Are there dependencies on this, and can it be changed?

- L461: You select CEs based on SPN output, which establishes plausibility in your results. Does this give your method an advantage over baselines? For example, on L484, you claim "unparalleled" plausibility.

- L471: You mention that the failure to terminate worsens the results. How pronounced is this effect?


Minor points:

- L82: "This work combines the tradition ..."—what tradition are you referring to? Please include a reference to clarify.

- Fig2: The heatmap legend is missing. Are the most likely points represented by the most yellow areas, particularly near 0 amount and 12 months?

- L252: "SPNs are a strict generalization"—this statement is missing a reference.

- L293: Typo: "the it"

- L376: Why do you need $v^{cont}$ and $v^{bin}$? Could you elaborate on the correspondence between these values and their respective roles in your approach? Does this correspondence offer advantages or properties?

- L468: What do you mean by "considering the differences between methods" when referring to finding the error acceptable?

- L497: What do you mean by "the limitations of all MIO methods"?

---

> ### Author Response · Authors · 2024-11-19
> **Rebuttal 1/3**
>
> We thank the Reviewer RaSc for their thorough review and pointing out a few typos. They have been fixed. We first address the reviewer's main concern, followed by other questions in the order they were asked, in 2 parts. Finally, we address minor questions in part 3.
>
> > Could you justify this assumption [that SPN provides the correct measure of plausibility]?
>
> We believe that SPNs are well suited for the problems, because (i) they naturally model distribution over continuous and discrete random variables; (ii) their simple formulation can be tightly approximated within MIO; (iii) and they are universal approximators (Nguyen & McLachlan, 2018).
>
> Other options are
>  - _Gaussian Mixture Models (GMMs)_ are designed only for continuous random variables. SPNs are a strict superset to GMMs.
>  - _Flow models_ are very flexible, but they model only distribution on continuous random variables. Since they are parametrized by neural networks, they might be rather difficult to formulate within MIO, especially considering their block nature relying on smooth non-linear functions (exp, softplus).
>  - _Neural auto-regressive models_ can model discrete and random variables, and they provide exact likelihood. But again, they use relatively large neural networks which might need non-linearities that are difficult to use within MIO (sigmoid, softmax).
>  - _Auto-encoders_ have, with respect to MIO, similar advantages and disadvantages as neural auto-regressive models. Furthermore, they provide only lower-bound estimates of true likelihood in the form of ELBO.
>
> We state the main advantages of SPNs on lines 82 (in Introduction) and 251 (in Related work) and we have added the above discussion to Appendix A.8.3.
>
> We are not aware of any thorough comparisons of SPNs to other models. Here, we list a few publications providing a comparison to some popular models.
>  - Comparison to Bayesian Networks and Markov models (Gens & Domingos, 2013) https://proceedings.mlr.press/v28/gens13.pdf
>  - The paper introducing SPTN where even the vanilla SPN is favorably compared to Masked auto-regressive flow models and GMMs. https://proceedings.mlr.press/v138/pevny20a.html
>  - Comparison to Hybrid Bayesian Networks (for mixed data)  https://cdn.aaai.org/ojs/11731/11731-13-15259-1-2-20201228.pdf
>  - Comparison to VAE https://ojs.aaai.org/index.php/AAAI/article/view/25883
>  - Comparison to Masked autoencoders (MADE) and VAE https://proceedings.mlr.press/v115/peharz20a/peharz20a.pdf
>  - Related Poisson-SPNs compared to Poisson Dependency Networks and Latent Dirichlet Allocation https://ojs.aaai.org/index.php/AAAI/article/view/10844
>
> Nguyen, H. D., & McLachlan, G. (2018). On approximations via convolution-defined mixture models. Communications in Statistics - Theory and Methods, 48(16), 3945–3955. https://doi.org/10.1080/03610926.2018.1487069
>
> > [...] provide an evaluation [...]?
>
> To empirically validate the quality of an SPN as a model of data likelihood, one would need to perform experiments on synthetic data from known underlying distributions. We agree that this would strengthen the paper and we are working on such experiments. If there is some distribution you suggest we would welcome it.
>
> > L53: What is the definition of an outlier in your context? Are you suggesting that an epsilon-ball around the factual could contain outliers [...]
>
> Throughout the paper, we use a common definition of an outlier as a sample with low likelihood (defined on L154). It is indeed true that an epsilon ball around a factual can contain outliers. This can happen when the probability density space is very anisotropic, i.e., it changes more dramatically with respect to some dimensions compared to others.
>
> > L80: Table 1 is not explained. Why are the "other desiderata" important?
>
> All columns of Table 1 are explained in the caption or, in the case of Plausibility and Sparsity, in the surrounding text. The discussion of Plausibility directly precedes L80 and Sparsity is discussed in the following example.
> Actionability, Plausibility, and Sparsity are also defined precisely in Section 2.1. The importance of the three desiderata is illustrated by the example in Figure 1 and its discussion. See L107 for a discussion of the importance of considering their combination, rather than solely plausibility.
>
> To reiterate, actionability is important since we aim for a counterfactual that can be realized. Sparsity is important because it is generally considered easier to make fewer changes. Plausibility is important because it is easier to make a change towards a value that is more likely. This likelihood is based on the fact that many other people (input vectors) have achieved a similar value. Model-agnostic methods are better since they can work with various predictor models. The ability to handle various non-continuous data (Complex data) well is especially important for tabular inputs. Finally, Exogenous generation is a great bonus, since CEs are not restricted to samples in the dataset.

---

> ### Author Response · Authors · 2024-11-19
> **Rebuttal 2/3**
>
> > L97-107: A definition of plausibility is missing in this section.
>
> This section describes Figure 1, showing the log-likelihood heatmap, which we link to plausibility on L68 and then indirectly again on L88 by stating that we use an SPN to measure plausibility. On L153, we define the plausibility of CEs as having high likelihood.
>
> > You claim that DiCE is the most plausible [...]
>
> The precise statement is “the most plausible explanation produced by DiCE”, meaning it is the one with the highest likelihood out of all CEs generated by DiCE (since DiCE generates a diverse set of CEs).
>
> > Is this [change in income] more "realizable" than borrowing less?
>
> The realizability of a certain CE depends on the specific context of an individual. It is non-trivial to assess which change is more realizable. Please note that the other methods require changes to at least 6 features, not solely borrowing less.
>
> > [...] could you provide more details on how these changes are measured?
>
> The number of feature changes is measured as the number of features for which the factual and counterfactual have different values. Mathematically, we could write this as $\|x^\prime - x\|_0$ (L150).
>
> > L184: Why do you take the median for categorical values?
>
> This section describes the encoding of mixed-value features (partly continuous and partly categorical). Since $x_j$ is the factual value, we would like the default continuous value ($F_j$) to be equal to it. But when $x_j$ takes a categorical value, we take $F_j$ to be the median value of all continuous values and set $d_j^{\mathrm{cont}} = 0$ so it does not affect the objective. We do this so that the counterfactual is least penalized for taking the median value when changing the value from a categorical to a continuous one. We have clarified the sentence in the paper.
>
> > L188: Why will at least one always be 0?
>
> If there was a solution with $l_j > 0$ and $u_j > 0$, we could obtain the same continuous value $c_j$ if we took $l_j^\prime = l_j - \min (l_j, u_j)$ and $u_j^\prime = u_j - \min (l_j, u_j)$. And because the solution with $l_j, u_j > 0$ would have a worse objective value (since we minimize the (weighted) sum $l_j + u_j$) it will not be returned as a solution.
>
> > L237: [outliers and plausibility not defined]
>
> Plausibility is defined on L153 in Section 2.1 as referenced on L238. We use the term outliers in accordance with the typical understanding, i.e., sample belonging to a low-likelihood region in the input space. This is stated as part of the plausibility definition on L154.
>
> > L260: Why do you work in log-space?
>
> The rationale is two-fold. Firstly, it enables the approximation of both sum and product nodes by linear constraints. Secondly, it makes the optimization less prone to numerical instabilities. We have added the rationale to the main body (L261-L263 in the updated document).
>
> > L332: [...] Are there dependencies on this, and can it be changed?
>
> There are no dependencies on this, it is an example of the threshold for the most typical case. It can be set arbitrarily, e.g., were the threshold at 0.5 we could set the constraint thresholds to $0.5 + \tau$ and $0.5 - \tau$. Further discussion on validity constraint is in Appendix A.3.1.
>
> > L461: [...] Does this [using SPNs in search] give your method an advantage over baselines?
>
> It is reasonable to assume that the use of SPN might give an advantage. However, notice that CEs from both DiCE and VAE undergo a second stage where we use the SPN to take the likeliest CE out of 10 generated. The results of those runs are on par with the other methods despite the use of an SPN.
>
> > L471: [...] How pronounced is this [terminating early] effect?
>
> This effect is shown in Table 3, where the success rate (of generating a valid CE) of LiCE (median) is shown for each dataset, the lowest being 55.6% on the GMSC dataset, and the highest 100% on the Credit dataset. The failure to generate a CE is caused by a combination of a tight time limit and a high lower bound on likelihood. See A.7.1 (and our answer to Reviewer Xi76) for more discussion.

---

> ### Author Response · Authors · 2024-11-19
> **Rebuttal 3/3  - Minor points**
>
> > L82: "This work combines the tradition ..."—what tradition are you referring to?
>
> Tradition here refers to the fields of research and their respective communities. Both fields (MIO and SPNs) are described in more detail in Prerequisites.
>
> > Fig2: Are the most likely points represented by the most yellow areas [...]?
>
> Indeed, _Figure 1_ displays log-likelihood where the brighter yellow color means higher likelihood.
>
> > L252: "SPNs are a strict generalization"—this statement is missing a reference.
>
> GMMs are weighted sums of Gaussian distributions. SPN with a single sum node is a direct model of a GMM, assuming that leaves contain Gaussian distributions. Aden-Ali & Ashtiani (2020) call GMM a “special case of SPNs”. We have added this reference.
>
> Aden-Ali, I. & Ashtiani, H.. (2020). On the Sample Complexity of Learning Sum-Product Networks. Proceedings of the Twenty Third International Conference on Artificial Intelligence and Statistics, in Proceedings of Machine Learning Research 108:4508-4518 Available from https://proceedings.mlr.press/v108/aden-ali20a.html.
>
> > L376: Why do you need $v^{cont}$ and $v^{bin}$?
>
> This notation is used to distinguish two parts of the weight vector $v$ since the value changes of categorical (binarized by one-hot encoding) and continuous variables are formulated differently. It is just a matter of notation to ease the writing of the objective function (Eq. 14).
>
> > L468: What do you mean by "considering the differences between methods" when referring to finding the error acceptable?
>
> This means that since the approximation error is notably lower than the difference in log-likelihood between LiCE and other CE methods (see Table 4), we find it to be an acceptable approximation. We have clarified the sentence.
>
> > L497: What do you mean by "the limitations of all MIO methods"?
>
> This refers to limitations in scalability and computational complexity. We added this specification to the paper.

---

> ### Comment · Reviewer_RaSc · 2024-11-20
>
> Thank you for your clarifications. While I appreciate your effort to address my concerns, I still have reservations about your definitions of plausibility and outliers, as well as the broader motivation of the paper. Specifically, the paper centers on improving plausibility and demonstrating that your method achieves this, but I find the problem to be ill-defined, and the experimentation appears biased towards your hypothesis (i.e., that your method outperforms state-of-the-art approaches).
>
> Definition of Outliers
>
> You stated:
>
> > We use the term outliers in accordance with the typical understanding, i.e., a sample belonging to a low-likelihood region in the input space. This is stated as part of the plausibility definition on L154.
>
> Could you provide a reference to support this "typical understanding"? For example, my understanding of the term aligns with the following definition:
>
> > "An outlying observation, or 'outlier,' is one that appears to deviate markedly from other members of the sample in which it occurs." [1]
>
> Your definition is based on likelihood, which fits your problem definition, and it's ok. I would still recommend being precise about it and not assuming a "typical understanding".
>
> Interconnection Between Actionability and Plausibility
>
> You wrote:
>
> > [...] actionability is important since we aim for a counterfactual that can be realized. Sparsity is important because it is generally considered easier to make fewer changes. Plausibility is important because it is easier to make a change towards a value that is more likely.
>
> My concern lies in the heavy interconnection between actionability and plausibility in your formulation. Actionability is defined as a set of constraints, which your method satisfies by design due to the use of MIO. Plausibility, meanwhile, is defined in terms of high likelihood, as estimated by your chosen method (SPNs). Since your method is built to satisfy these definitions by construction, the evaluation appears inherently biased towards your approach.
>
> To address this, I recommend generating synthetic data from known distributions, applying constraints to the data-generating process (DGP), and then fitting the SPN and other baseline methods. This would allow for a more robust and unbiased evaluation of plausibility and actionability.
>
> Reference
> [1] Grubbs, F. E. (1969). "Procedures for detecting outlying observations in samples." Technometrics, 11(1), 1–21.

---

> > ### Author Response · Authors · 2024-11-20
> >
> > We consider an outlier to be a sample which occurs with such a low probability that it might raise a suspicion of being generated by a different distribution, which fits the definition [1]. We agree that this should be made precise, especially with respect to the fact that some definition of outliers are based on distances, i.e. outlier is a sample which is far from the other. Hence, a definition make this precise.
> >
> > After our discussion, we now better understand why the lack of precise definition might lead to confusion between plausibility and actionability. An event, which might be plausible might not be actionable. In the running example of loans, it can happen that decreasing the age of applicant can increase plausibility, but it is not actionable because we just cannot make ourselves younger. Therefore actionability enforces constraints.
> >
> > We are working on synthetic example, though we are not sure we understand clearly what do you mean by "applying constraints to the data-generating process"? Possibly, if you have some synthetic dataset in mind, can you share it? We are happy to test the methods on it.

---

> > > ### Comment · Reviewer_RaSc · 2024-11-20
> > >
> > > Thank you for your clarification and example. I agree with your interpretation of the interconnection between actionability and plausibility, and I appreciate the effort you’ve put into explaining your perspective.
> > >
> > > I apologize for the lack of clarity in my earlier suggestion regarding synthetic experiments. The constraints I mentioned differ from those you refer to as satisfying actionability. While your constraints focus on the CEs and embody your definition of actionability, the constraints I was referring to are inherent to the DGP itself.
> > >
> > > The constraints I had in mind pertain to the functional forms (or conditional probabilities) governing the relationships in the DGP. For example, a simple functional form could be $X_3 = X_1+X_2$. A straightforward way to generate pseudo-real data with such structure is by employing a Bayesian Network (BN). Using a BN, you can directly read the conditional probabilities from the conditional probability tables (CPTs), which makes it straightforward to encode the dependencies in the DGP. Using BNs would also make the validation of SPN easier since the joint distribution of the data is defined by the CPTs. The bnlearn repository (https://www.bnlearn.com/bnrepository/) contains many examples of Bayesian Networks that could be used for this purpose. For reference, [2] demonstrates the validation of counterfactual explanations on some datasets from this repository.
> > >
> > > Reference
> > > [2] Smyth, Barry, and Mark T. Keane. "A few good counterfactuals: generating interpretable, plausible and diverse counterfactual explanations." International Conference on Case-Based Reasoning. Cham: Springer International Publishing, 2022.

---

> > > > ### Author Response · Authors · 2024-11-21
> > > >
> > > > Thank you for the clarification about the experiment setup. We chose the `bnlearn` Python library (https://github.com/erdogant/bnlearn/) to model the Bayesian Networks (BNs).
> > > >
> > > > We ran synthetic experiments for 8 Bayesian Networks (7 from the suggested bnrepository) of varying sizes in the following setting: We sample 10000 points using the BN, fit an SPN on the data and then sample 1000 new points for which we estimate log-likelihood and compute log-probability from the Bayesian Network. We then compute the (Pearson's) correlation coefficient and Total Variation (for the smaller-sized BNs).
> > > >
> > > > Below, we show the results when taking the best-fitted SPN out of 5 seeded runs of the above setup for each Bayesian Network.
> > > >  |                         | sprinkler | asia   | sachs | child | water | alarm | win95pts | andes |
> > > > |-------------------------|-----------|--------|-------|-------|-------|-------|----------|-------|
> > > >  | # of features (nodes)   | 4         | 8      | 11    | 20    | 32    | 37    | 76       | 223   |
> > > >  | Correlation coefficient | 0.996     | 0.990  | 0.964 | 0.954 | 0.955 | 0.959 | 0.959    | 0.793 |
> > > >  | Total Variation         | 0.0168    | 0.0730 | -     | -     | -     | -     | -        | -     |
> > > >
> > > > The two measures correlate rather closely. The quality of the fit drops for the biggest BN. This might be partly caused by 10 000 samples being  too few for that many features.

---

> > > > > ### Author Response · Authors · 2024-11-22
> > > > > **Empirical evaulation of SPN as a model of data distribution**
> > > > >
> > > > > We have added the above experiments to the Appendix, with extended description and discussion. It is Section A.8.4 in the new version of the paper.

---

> > > > > > ### Comment · Reviewer_RaSc · 2024-11-22
> > > > > >
> > > > > > Thank you for providing results regarding the capability of SPNs to model structured DGPs.
> > > > > >
> > > > > > While I do not agree with relying solely on correlation coefficients, I understand the need to summarize results given the constraints, and I appreciate that a more in-depth analysis may not be feasible within the given time frame. That said, a metric better suited for evaluating non-parametric distribution similarity, such as Bhattacharyya distance, Hellinger distance, or KL divergence, would likely provide a more meaningful assessment in this context. Additionally, introducing a reasonable baseline (e.g., GMMs, VAEs, KDE, or MC methods) would be necessary to assess whether SPNs effectively approximate the underlying distribution.
> > > > > >
> > > > > > More importantly, the results do not alleviate my concerns about potential bias in your evaluation, as SPNs are being used as the ground truth for plausibility. This creates a situation where any errors made by the SPN inherently favor your method, and there is no information on how often or how significantly the SPN may have been incorrect in the experiments. To substantiate your claim that your method produces more plausible CEs than the baselines, it would be necessary to evaluate plausibility independently of SPNs as the ground truth. For instance, this could involve using the data from the BNs and using the probabilities from the CPTs as ground truth. While I acknowledge that this may be challenging, such an evaluation would, in my view, considerably strengthen the validity of your conclusions.

---

> > > > > > > ### Author Response · Authors · 2024-11-23
> > > > > > >
> > > > > > > We see your point and would like to comment little bit.
> > > > > > > * Correlation is not the best, but a strong correlation demonstrates the values of SPN are indicative of true values. This means that if we optimize with respect to lkl provided by SPN, we are indirectly optimizing true value.
> > > > > > > * BN you have suggested seems to use to be reasonable baseline. Which other model than BN, which we have validated on artificial data, would be reasonable baseline? GMM works only for continuous random variables and it is a subset of SPNs. VAE does not provide exact likelihood. KDE to my knowledge are used only for continous random variables. We do not know, what do you mean by MC.
> > > > > > >
> > > > > > > So while having other model would be nice, I think you suggestion to use BN and compare to it was a good solution.

---

> > > > > > > > ### Author Response · Authors · 2024-11-25
> > > > > > > > **Ad concerns about evaluation bias**
> > > > > > > >
> > > > > > > > To alleviate your concerns about potential bias in our evaluation, we performed another set of experiments. In an ideal scenario, we would evaluate the true probability (density) of a given sample. We, unfortunately, do not have access to that distribution. In the place of that evaluation, we do the following:
> > > > > > > >
> > > > > > > > We utilize 3 of the above-used Bayesian Networks of varying size (asia, alarm, and win95pts), choose a target variable (dysp, BP, and Problem1, respectively), and sample a training dataset of 10,000 samples. On this training dataset, we train an SPN and a Neural Network model that we utilize to generate counterfactuals for 100 factuals. We perform this for 5 different seeds for each BN.
> > > > > > > >
> > > > > > > > Results are in Appendix A.8.5 in Tables 15, 16, and 17. We have also added a paragraph summarizing our fruitful discussion in the limitations paragraph.
> > > > > > > >
> > > > > > > > The results show that LiCE methods, especially the likelihood optimizing variant ($\alpha = 1$), perform comparably to other methods even when taking into account the true distribution.
> > > > > > > >
> > > > > > > > Additionally, note that:
> > > > > > > >  - performance improvements in terms of distance and sparsity reflect experiments on real data;
> > > > > > > >  - only MIO-based methods generate 100% of valid counterfactuals, other methods generate 55.8% at best;
> > > > > > > >  - the time complexity of LiCE is on par with other methods;
> > > > > > > >  - while not perfect, SPN likelihood generally correlates with the true probability (as discussed above).
> > > > > > > >
> > > > > > > > We omit the results of DiCE and VAE methods due to them not finishing in time.

---

> > > > > > > > > ### Comment · Reviewer_RaSc · 2024-11-27
> > > > > > > > >
> > > > > > > > > Thank you for providing additional results to address my concerns about the evaluation of plausibility.
> > > > > > > > >
> > > > > > > > > I have a few comments regarding the additional experiments. Please note that I am not asking for further experiments at this stage, but rather suggesting ways to improve the presentation and, if possible, the rigor of your results:
> > > > > > > > >
> > > > > > > > > - Why does the true probability change across different methods? This seems counterintuitive and warrants further explanation.
> > > > > > > > > - What similarity metric are you using? You mention a distance—does this directly measure plausibility?
> > > > > > > > > - In the Asia dataset, there do not appear to be significant differences in similarity (assuming this metric represents plausibility). However, there are notable differences for the Alarm and win95pts datasets. What about the other datasets? Andes distribution seems quite off for example. Understanding and explaining these differences across datasets would help justify the effectiveness of your method. Additionally, applying statistical tests to demonstrate that these differences are significant would add rigor to your results.
> > > > > > > > >
> > > > > > > > > More broadly, I would like to emphasize that performing comparably to other methods is very different from being the best and always 100% plausible, as claimed in the main manuscript. These results, once refined and better analyzed, should be included in the main manuscript rather than relegated to the appendix. Furthermore, the overall message of the paper should be adjusted to more accurately reflect these findings.

---

> > > > > > > > > > ### Author Response · Authors · 2024-11-27
> > > > > > > > > >
> > > > > > > > > > Thank you for your further comments, let us address your points one by one:
> > > > > > > > > >
> > > > > > > > > > - The true probability changes because it is the probability of the generated counterfactuals. Indeed, each method generates different counterfactuals for the same factuals. If the counterfactuals were the same, SPN would also evaluate their likelihood as the same, and thus, the results would not differ between CE methods.
> > > > > > > > > > - Similarity (and sparsity) are defined the same as in the main body of the paper (Table 4), i.e., the similarity is a distance from the factual to counterfactual. The measures are shown to enable a deeper comparison of the results. We also argue that similarity and sparsity are related to the "usefulness" of a CE for the end user, which is one of the motivations behind studying the plausibility of CEs. To evaluate plausibility, defined as CE having high likelihood, the relevant columns in Tables 15-17 are the "Log-likelihood estimate" (given by the SPN) and "True probability" (evaluated using the BNs).
> > > > > > > > > > - We test generating the CEs using one small, one medium, and one large Bayesian Network to cover a variety of input sizes. The inaccuracies of SPN regarding the Andes BN can be caused by a multitude of reasons, one of which could be undersampling, given that there are 2^223 possible values, and we sample only 10,000 (< 2^14) points. We will perform statistical tests and include their results in the paper.
> > > > > > > > > >
> > > > > > > > > > While we never claimed 100% plausibility anywhere in the paper (the percentages refer to the validity or actionability of CEs), we agree that the claim on line 489 might've suggested a stronger notion of plausibility than the one we measured. We changed it to "The plausibility results of LiCE (median), evaluated by the SPN, seem to be dominant." We have already added a summarization of the evaluated limitations of using an SPN to estimate plausibility on lines 507-511. We welcome concrete suggestions regarding what might be missing.

---

> > > > > > > > > > > ### Author Response · Authors · 2024-11-28
> > > > > > > > > > > **Statistical tests**
> > > > > > > > > > >
> > > > > > > > > > > We have tested the statistical significance of the method comparison by ranking the true probability of generated CEs. Assuming that the failure to generate a valid CE should rank lower than any valid CE, we show that our method performs significantly better.
> > > > > > > > > > >
> > > > > > > > > > > If we instead generously consider only factuals for which all methods generated a valid CE, the image becomes less clear. On small and medium-sized BNs, LiCE still ranks the highest. However, on the large BN (win95pts), the proposed methods score poorly. These results suggest a dependence on the quality of the trained SPN, which, in turn, depends on the amount of samples provided, given the size of the input space. The significantly better models are FACE and PROPLACE, which use the training data directly. It is also possible that the performance could be improved by using a different training algorithm for the SPN.
> > > > > > > > > > >
> > > > > > > > > > > We have added the statistical tests with results and discussion to Section A.8.5 in the paper.

---

### Official Review · Reviewer_Xi76 · 2024-11-04

**Soundness:** 4
**Presentation:** 3
**Contribution:** 4
**Rating:** 8
**Confidence:** 4

**Summary:**

This paper deals with the task of finding counterfactual explanations (x’) for a multi-class classification model (h(x)). Specifically, the authors aim to find a counterfactual set (C) satisfying desiderata specified in Guidotti (2022). i.e. apart from being a valid counterfactual, each x’ \in C must be similar to the original example x, involve changing as few features as possible (sparse), comply with domain constraints such as monotonicity of age (actionable), and not be an outlier (plausibility). Additionally, C must be as diverse as possible and follow causal domain knowledge. The authors approach this problem from a mixed integer optimization perspective and use sum-product networks to model the data distribution (P(X)). To this effect, they translate the desiderata into constraints and solve the optimization problem using the OMLT library (Ceccon et al., 2022). While the trained SPN cannot be encoded exactly in log-space, the authors develop an approximate encoding and bound the likelihood. The authors call their proposed counterfactual generation approach Likely Counterfactual Explanations (LiCE) and define two variants of LiCE, one based on likelihood (upper) threshold of train-set median (LiCE (median)) and the other on minimizing a linear combination of distance and likelihood (LiCE (optimize)). They evaluate the two variants by comparing them against a non-SPN variant (MIO) and on prior work including DiCE (Mothilal et al., 2020), C-CHVAE (Pawelczyk et al., 2020), FACE (Poyiadzi et al., 2020) and PROPLACE (Jiang et al., 2024) on 3 financial data sets focusing on plausibility (as measured by log-likelihood), similarity (counterfactual distance), and sparsity (number of modified features). Their experiments show that both LiCE variants outperform the baselines, and while LiCE (median) excels at plausibility of generated counterfactual sets, LiCE (optimize) is the best at similarity even beating MIO.

**Strengths:**

- LiCE is a principled method of inferring *plausible* counterfactual explanations that satisfy several desiderata. The MIO approach is flexible enough to accommodate additional criteria.
- The MIO formulation for SPN inference is itself a significant contribution. It opens up space for work on SPN inference tasks such as finding the entire Maximum a Posterori (MAP) set possible. Existing work has focused on finding single solutions (e.g., Poon and Domingos, 2011 and Arya et al., 2024).

Arya, Shivvrat, Tahrima Rahman, and Vibhav Gogate. "Neural Network Approximators for Marginal MAP in Probabilistic Circuits." AAAI 2024.

**Weaknesses:**

The Accuracy of likelihood assessment is limited by the expressivity of SPNs. This might be at least partially resolved by using PFCs (Sidheekh et al., 2023) to improve performance on high-dimensional domains,

Sidheekh, Sahil, Kristian Kersting, and Sriraam Natarajan. "Probabilistic flow circuits: towards unified deep models for tractable probabilistic inference." UAI 2023.

**Questions:**

- Would making structural assumptions (e.g., determinism) about SPNs make it easier to encode exactly?
- Can you elaborate on the setup and significance of the VAE baseline from Mahajan et al. (2020)?
- The results mention that LICE (median) times out for some of the cases, and the appendix mentions that the time limit for each run was 2 minutes. Are these results sensitive to the time limit duration? Would they change drastically for a small increase in the time limit?

---

> ### Author Response · Authors · 2024-11-19
> **Rebuttal**
>
> We thank the reviewer Xi76 for their review and address their questions below:
>
> > [...] limited expressivity of SPNs [...] might be at least partially resolved by using PFCs [...]
>
> Using Probabilistic flow circuits is an interesting idea. It would be possible if the one-dimensional flow models in leaves were discretized. This approximation would be possible with controlled error since the discretization is done after fitting.
>
> > Would making structural assumptions (e.g., determinism) about SPNs make it easier to encode exactly?
>
> The approximation requirement stems from the fact that sum nodes are computed by $\log \sum \exp( \cdot )$ function in the log domain (which we need for the product nodes to be computed using sums). We approximate $\log \sum \exp( \cdot )$ by $\max$ function that can be represented by linear constraints within MIO, arriving at mixed-integer linear formulation. We show empirically in Table 2, that the approximation is rather tight.
>
> > Can you elaborate on the setup and significance of the VAE baseline from Mahajan et al. (2020)?
>
> Indeed, the VAE method scores poorly on the GMSC dataset, regarding the number of successful CE generations. This could be partly caused by the inability to learn a good VAE with the resources and hyperparameters provided. The implementation does not provide default parameters for learning configuration so we fix them to values that worked reasonably well for the Adult dataset on which initial tests were made. The specific configuration is in the supplementary material in `compute_CEs_py311.py` script on lines 107-114. Though the results of VAE could possibly be improved by finding a more suitable hyperparameter setup, the variance in performance on different datasets using the same configuration is interesting.
>
> > [...] Are these results sensitive to the time limit duration?
>
> The 2-minute time limit certainly plays a role, but it might also be the case that the bound on the likelihood is too tight. A less restrictive bound can finish within the time limit. See the LiCE (quartile) discussion in A.7.1 and the time comparison in Table 12.
>
> > Would they change drastically for a small increase in the time limit?
>
> By increasing the time limit from 2 to 5 minutes, LiCE (median) finds a valid CE for 64.8% of factuals from the GMSC dataset (almost 10 percentage points increase) and 91.6% of factuals from the Adult dataset (increase by 0.4 p.p.). This suggests that the lower bound on likelihood might be quite restrictive when set to the median value.

---

> > ### Comment · Reviewer_Xi76 · 2024-11-26
> >
> > Thank you for the detailed response. I will keep my positive evaluation

---

### Meta-Review · Area_Chair_QRsN · 2024-12-18

**Metareview:**

This paper is truly in the borderline where two reviewers view this positively while two are others are a bit negative about this. All the reviewers agree that the integration of SPNs and MIO to optimize multiple constraints is quite novel. The idea of learning SPNs and then finding a counterfactual set is quite interesting. There were some concerns about the computational complexity, comparison to baselines and  better experimentation could improve the paper.

**Additional Comments On Reviewer Discussion:**

The reviewers (particularly the negative score reviewers) did a great job of interacting with the authors. Most of the concerns seemed to have been addressed. There is some issue about computational complexity by reviewer kD9A and empirical evaluations by RaSC but I feel that the authors have more or less addressed many of these issues in the response.

---

### Decision · Program_Chairs · 2025-01-22

Accept (Poster)